PREPARED FOR SUBMISSION TO SCIPOST

# Trans-series from condensates

**Marcos Mariño**[a] **and Ramon Miravitllas**[b]

[a]*Département de Physique Théorique et Section de Mathématiques*
*Université de Genève, Genève, CH-1211 Switzerland*

[b]*HUN-REN Wigner Research Centre for Physics*
*Konkoly-Thege Miklós u. 29-33, 1121 Budapest, Hungary*

*E-mail:* Marcos.Marino@unige.ch, ramon.miravitllas.mas@wigner.hu

ABSTRACT: The Shifman–Vainshtein–Zakharov (SVZ) sum rules provide a method to obtain trans-series expansions in many quantum field theories, in which exponentially small corrections are calculated by combining the operator product expansion with the assumption of vacuum condensates. In some solvable models, exact expressions for trans-series can be obtained from non-perturbative results, and this makes it possible to test the SVZ method by comparing its predictions to these exact trans-series. In this paper we perform such a precision test in the example of the fermion self-energy in the Gross–Neveu model. Its exact trans-series expansion can be extracted from the large $N$ solution, at the first non-trivial order in $1/N$. It is given by an infinite series of exponentially small corrections involving factorially divergent power series in the 't Hooft parameter. We show that the first two corrections are associated to two-quark and four-quark condensates, and we reproduce the corresponding power series exactly, and at all loops, by using the SVZ method. In addition, the numerical values of the condensates can be extracted from the exact result, up to order $1/N$.

## 1 Introduction

In quantum field theory (QFT), perturbative series give the asymptotic expansion of observables at small coupling. There are many indications that this expansion can be upgraded to a *trans-series*, i.e. a generalization of the perturbative series which includes exponentially small corrections, in such a way that the exact value of the observable can be obtained by an appropriate resummation.

Examples in QFT where we know explicitly the detailed structure of the trans-series are scarce. In some two-dimensional asymptotically free theories, one can compute certain observables exactly, and then show that this exact expression can be re-expressed or "decoded" as a resummed trans-series. A beautiful realization of this idea was achieved in [1], building on previous work [2, 3], in the case of the two-point function of the non-linear sigma model at next-to-leading order in the $1/N$ expansion[1]. More recently, the free energy of integrable models coupled to a conserved charge was decoded as a resummed trans-series, even at finite $N$ [4–7].

In generic QFTs, in which no analytic answer is known for the observables, physicists have devised two ways of upgrading the perturbative series to a trans-series. The first one is to add instanton corrections, coming from non-trivial saddle-points of the path integral. Instanton

---

[1]The $1/N$ expansion, when used appropriately, gives a power series in $1/N$ where each term is an exact, non-perturbative function of the renormalized 't Hooft coupling, and not just a formal power series thereof. Sometimes we will use the expression "exact results at large $N$," or similar ones, to refer to this type of non-perturbative functions.

calculus is plagued with problems and it is fair to say that it is of limited use, except in super-symmetric theories or in very simple models. The second method to obtain trans-series can be applied to correlation functions in very general QFTs. It combines Wilson's operator product expansion (OPE) with some assumptions on the vacuum structure of the theory[2]. More precisely, this method assumes the existence of *condensates*, or non-zero vacuum expectation values (vevs), for the operators appearing in the OPE. It was used by Politzer in [10] to calculate the quark propagator in QCD beyond perturbation theory, and then extended and systematized in the famous QCD sum rules of Shifman, Vainshtein and Zakharov (SVZ) in [11, 12]

An obvious question is whether the method of OPE and vacuum condensates provides the correct trans-series representation of correlation functions. Although this might be obvious to most practitioners in the field, there are various reasons for a detailed inquiry. For example, it could be the case that the OPE provides only an approximate parametrization of non-perturbative corrections, rather than the real thing. There has also been some debate concerning which form of the OPE should be used to calculate trans-series. Most of the sum rule calculations done by physicists are based on a simplified or "practical" version of the OPE, in which the Wilson coefficients are calculated perturbatively, while the vevs of the operators contain the non-perturbative information, but it has been suggested [13, 14] that one might need more complicated procedures. For us, another important motivation to revisit these questions was the discovery, in some integrable models, of fractional power corrections which can not be easily accommodated in the current OPE/renormalon paradigm [4].

It is a good idea to ask foundational questions in simpler, solvable models where they can be answered with precision. In the case of the SVZ method this has been done in various papers, starting in the 1980s [1–3, 13–15]. These works usually focused on the two-dimensional non-linear sigma model at large $N$, and they extracted trans-series from exact results at leading and next-to-leading order in the $1/N$ expansion, as we mentioned above. However, and somewhat surprisingly, many of these studies essentially *assumed* that the trans-series obtained from the exact result could be obtained from an OPE calculation with condensates, but did not verify it explicitly. Some direct tests of the SVZ method were performed in [13–15] but in rather simple situations. In particular, the non-trivial perturbative series attached to the power corrections of [1] in the non-linear sigma model were never reproduced by a direct OPE calculation.

The goal of this paper is to provide such a direct comparison between an exact trans-series and a standard calculation of power corrections. The original example of the two-dimensional non-linear sigma model is not the simplest one to perform such a comparison, and we focus instead on a fermionic cousin, the Gross–Neveu (GN) model [16], where trans-series are of comparable complexity. The self-energy of elementary fermions (or "quarks") in this model can be calculated at the first non-trivial order in the $1/N$ expansion, as an exact function of the external momentum and the mass gap [17]. By using the Mellin transform techniques of [1] one can obtain an explicit trans-series representation of this function, involving an infinite series of power corrections. Schematically, we have

$$\Sigma(p) = \not{p}\Sigma_0(\lambda) + \Lambda\Sigma_1(\lambda) + \not{p}\frac{\Lambda^2}{p^2}\Sigma_2(\lambda) + \cdots , \tag{1.1}$$

where $\Lambda$ is the dynamically generated scale and $\lambda$ is the 't Hooft coupling. The series $\Sigma_0(\lambda)$ is the perturbative series, but each power correction involves a factorially divergent series $\Sigma_n(\lambda)$,

---

[2]The fact that the method of OPE with condensates leads to trans-series, in the sense of the theory of resurgence, was pointed out some time ago in [8]. See also [9] for related observations.

$n = 1, 2, \cdots$. If the OPE picture is correct, one should be able to reproduce these series by doing perturbation theory in the background of the appropriate vacuum condensates. This is precisely what we verify with complete success, and at *all* loops, for the first two power corrections, which are associated to the two-quark condensate and to the four-quark condensate (these are the terms with $n = 1, 2$ in the equation above, respectively). Once this is done, the values of the condensates at next-to-leading order (NLO) in $1/N$, which are unknown parameters in the sum rules, can be extracted from the large $N$ result.

Our calculation provides a precise and direct test of the SVZ method in the Gross–Neveu model, at the first non-trivial order in $1/N$ expansion, and it illustrates various conceptual and practical issues of the method. For example, it shows that the four-quark condensate is not ambiguous at leading order in the $1/N$ expansion, due to factorization, but it is indeed ambiguous at subleading order, as expected from the results of [2, 3]. Our calculation is done with the "practical" version of the OPE, which leads to the correct result in this example. An amusing spin-off result of this work is a diagrammatic derivation of the beta function of the model, at next-to-leading order in the $1/N$, which seems to be simpler than the approach usually followed in the literature [18, 19].

We should mention that the paper [20] considered the chiral Gross–Neveu model, and compared the exact $1/N$ result for the propagator of the sigma particle to an OPE calculation with condensates, to leading order in the 't Hooft coupling. The resulting trans-series is much simpler than the ones considered here. We derive this result for the sigma propagator, in the Gross–Neveu model, at the end of section 4.4.

This paper is structured as follows. In section 2 we review or derive various basic results for the GN model which will be needed for the paper. In section 3 we obtain the trans-series expression for the exact two-point function at the first non-trivial order in the $1/N$ expansion. In section 4 we calculate the two-point function in perturbation theory with condensates, where we include the first two power corrections, and we reproduce exactly the trans-series derived from the $1/N$ expansion. Finally, in 5 we conclude and present some questions and open problems. The Appendix collects some diagrammatic tools. It summarizes an important technique to calculate all-loop results in the $1/N$ expansion, due to Palanques-Mestre and Pascual [18, 21], which is used throughout the paper.

## 2 The Gross–Neveu model

The GN model is a two dimensional QFT, involving an $N$-uple of Dirac fermions with a quartic interaction, which was introduced in [16] as a toy model for various important physical phenomena. First of all, the GN model is asymptotically free. It can be solved in the large $N$ limit, where it can be shown that quantum effects lead to spontaneous symmetry breaking of a discrete $\mathbb{Z}_2$ symmetry and the formation of a bilinear fermion condensate. For these reasons, the GN model can be seen as a toy model for the quark sector of QCD. In addition, the model is integrable at the quantum level, its exact $S$-matrix has been conjectured in [22], and its spectrum is extremely rich. In this section we will review some aspects of the model which we will need in our precision test of the SVZ method.

We will work in Minkowski space, and our choice of Dirac algebra in two dimensions is:

$$\gamma^0 = \sigma^2, \qquad \gamma^1 = \mathrm{i}\sigma^1, \qquad \gamma^5 = \sigma^3. \tag{2.1}$$

The Lagrangian density describing the theory is

$$\mathcal{L} = \mathrm{i}\overline{\boldsymbol{\psi}} \cdot \boldsymbol{\partial}\!\!\!/\boldsymbol{\psi} + \frac{g_0}{2}\big(\overline{\boldsymbol{\psi}} \cdot \boldsymbol{\psi}\big)^2, \tag{2.2}$$

where $\boldsymbol{\psi} = (\psi_1, \cdots, \psi_N)$ is an $N$-uple of Dirac fermions. The model has a continuous $U(N)$ global symmetry, and a $\mathbb{Z}_2$ discrete symmetry

$$\boldsymbol{\psi} \to \gamma_5 \boldsymbol{\psi}. \tag{2.3}$$

In order to keep track of large $N$ counting, it is extremely useful to introduce an auxiliary scalar field $\sigma$ and write the GN Lagrangian as

$$\mathcal{L}_\sigma = \mathrm{i}\overline{\boldsymbol{\psi}} \cdot \slashed{\partial}\boldsymbol{\psi} - \frac{1}{2}\sigma^2 + \sqrt{g_0}\sigma\overline{\boldsymbol{\psi}} \cdot \boldsymbol{\psi}. \tag{2.4}$$

The original Lagrangian is obtained by integrating out $\sigma$. The symmetry (2.3) reads now

$$\boldsymbol{\psi} \to \gamma_5 \boldsymbol{\psi}, \qquad \sigma \to -\sigma. \tag{2.5}$$

One can also add a bare mass term for the fermions of the form

$$\mathcal{L}_{m_f} = -m_{0f}\overline{\boldsymbol{\psi}}\boldsymbol{\psi}, \tag{2.6}$$

although we will consider the massless theory with $m_{0f} = 0$ (as is well-known [16], even when $m_{0f} = 0$ a dynamical mass is generated at the quantum level and can be calculated in the $1/N$ expansion). From now one we will work with the Lagrangian (2.4). To write down the Feynman diagrams, we represent fermions by continuous lines and sigma particles by dashed lines. The fermion propagator in momentum space is given by

$$S_0(p)_{ij}^{\alpha\beta} = \left(\frac{\mathrm{i}}{\slashed{p} - m_{0f}}\right)^{\alpha\beta} \delta_{ij}. \tag{2.7}$$

(Latin sub-indices are $U(N)$ indices, while Greek super-indices are spinor indices.) The propagator of the $\sigma$ field is $-\mathrm{i}$, and there is a single interaction vertex $\mathrm{i}\sqrt{g_0}$.

The GN model is renormalizable and asymptotically free. We will almost always adopt the $\overline{\text{MS}}$ scheme and mostly work with bare fields, which we will simply denote by $\boldsymbol{\psi}$. Renormalized fields will be denoted by $\boldsymbol{\psi}_R$. The renormalization constants are defined as usual by

$$\boldsymbol{\psi} = Z_\psi^{1/2}\boldsymbol{\psi}_R, \qquad g_0 = (\nu^2)^{\epsilon/2}Z_g g, \qquad m_{0f} = Z_m m_f. \tag{2.8}$$

Our convention for $\epsilon$ is

$$d = 2 - \epsilon \tag{2.9}$$

and

$$\nu^2 = \mu^2 \mathrm{e}^{\gamma_E - \log(4\pi)}, \tag{2.10}$$

where $d$ is the number of space-time dimensions in dimensional regularization and $\mu$ is the scale parameter. The beta function is defined as

$$\beta(g; \epsilon) = -\frac{\epsilon g}{1 + g\partial_g \log Z_g} = -\epsilon g + \beta(g), \tag{2.11}$$

with

$$\beta(g) = -\sum_{k \geq 0} \beta_k g^{k+2}, \tag{2.12}$$

while
$$\gamma(g) = \beta(g;\epsilon)\frac{\partial \log Z_\psi}{\partial g}, \qquad \gamma_m(g) = \beta(g;\epsilon)\frac{\partial \log Z_m}{\partial g} \tag{2.13}$$

are the anomalous dimension of the field and the mass, respectively. The renormalization functions are known to four loops in conventional perturbation theory, see [23] for recent results and references to the literature. However, we will work in the $1/N$ expansion. We define the 't Hooft parameter
$$\lambda = \frac{gN}{\pi} \tag{2.14}$$

whose beta function is
$$\beta_\lambda(\lambda;\epsilon) = \frac{N}{\pi}\beta(g;\epsilon) = -\epsilon\lambda + \beta_\lambda(\lambda). \tag{2.15}$$

The function $\beta_\lambda(\lambda)$ has a $1/N$ expansion at fixed 't Hooft coupling given by
$$\beta_\lambda(\lambda) = \sum_{j\geq 0}\beta_\lambda^{(j)}(\lambda)N^{-j} \tag{2.16}$$

and similarly for $\beta_\lambda(\lambda;\epsilon)$. We note that
$$\beta_\lambda^{(0)}(\lambda;\epsilon) = -\epsilon\lambda - \lambda^2, \qquad \beta_\lambda^{(j)}(\lambda;\epsilon) = \beta_\lambda^{(j)}(\lambda), \quad j \geq 1. \tag{2.17}$$

The first correction $\beta_\lambda^{(1)}(\lambda)$ is known in closed form [24] and is given by
$$\beta_\lambda^{(1)}(\lambda) = \lambda^2\left(1 + \int_0^\lambda \frac{\Gamma(2+u)}{(2+u)\Gamma^3\left(1+\frac{u}{2}\right)\Gamma\left(1-\frac{u}{2}\right)}\mathrm{d}u\right). \tag{2.18}$$

We have similar results for the anomalous dimensions. The mass anomalous dimension has a $1/N$ expansion of the form
$$\gamma_m(\lambda) = \sum_{j\geq 0}\gamma_m^{(j)}(\lambda)N^{-j}, \tag{2.19}$$

where
$$\gamma_m^{(0)}(\lambda) = \lambda, \tag{2.20}$$

and the first non-trivial correction is given by [25]
$$\gamma_m^{(1)}(\lambda) = \chi(\lambda) - \frac{\beta_\lambda^{(1)}(\lambda)}{\lambda} \tag{2.21}$$

where
$$\chi(\lambda) = \frac{\lambda\Gamma(2+\lambda)}{(2+\lambda)\Gamma^3\left(1+\frac{\lambda}{2}\right)\Gamma\left(1-\frac{\lambda}{2}\right)}. \tag{2.22}$$

Finally, the field anomalous dimension has the $1/N$ expansion
$$\gamma(\lambda) = \sum_{j\geq 1}\gamma^{(j)}(\lambda)N^{-j}, \tag{2.23}$$

where
$$\gamma^{(1)}(\lambda) = \frac{\lambda^2}{2}\frac{1}{2+\lambda}\frac{\Gamma(1+\lambda)}{\Gamma^3\left(1+\frac{\lambda}{2}\right)\Gamma\left(1-\frac{\lambda}{2}\right)}. \tag{2.24}$$

As we will see, the functions (2.24), (2.22) and (2.18) will be obtained as spin-offs of our trans-series calculation for the perturbative, the two-quark condensate, and the four-quark condensate sectors, respectively. The renormalization constants in (2.8) can be recovered from the renormalization functions. We have, for the coupling constant,

$$Z_g = \exp\left[-\int_0^g \frac{\mathrm{d}u}{u} \frac{\beta(u)}{\beta(u;\epsilon)}\right], \tag{2.25}$$

while, for the field and mass renormalization, one finds

$$Z_\psi = \exp\left[\int_0^g \mathrm{d}u \frac{\gamma(u)}{\beta(u;\epsilon)}\right], \qquad Z_m = \exp\left[\int_0^g \mathrm{d}u \frac{\gamma_m(u)}{\beta(u;\epsilon)}\right]. \tag{2.26}$$

The renormalization constants can also be obtained in a $1/N$ expansion by simply re-expressing everything in terms of the 't Hooft coupling. In particular, this coupling renormalizes as

$$\lambda_0 = (\nu^2)^{\epsilon/2} Z_\lambda \lambda, \tag{2.27}$$

where $Z_\lambda$ is the renormalization constant $Z_g$ expressed in terms of $\lambda$ and organized in a $1/N$ expansion. It is given explicitly as

$$Z_\lambda = \exp\left[-\int_0^\lambda \frac{\mathrm{d}u}{u} \frac{\beta_\lambda(u)}{\beta_\lambda(u;\epsilon)}\right]. \tag{2.28}$$

We will also need to renormalize the composite operators appearing in the OPE. We will denote renormalized composite operators by a bracket, $[\mathcal{O}]$. The renormalization constants are defined by

$$\mathcal{O}_i = \mathsf{Z}_{ij}[\mathcal{O}_j], \tag{2.29}$$

where repeated indices are summed, and we have assumed mixing between a set of operators $[\mathcal{O}_i]$, $i = 1, \cdots, n$. Our convention for the matrix of anomalous dimensions is

$$\boldsymbol{\gamma} = -\beta(g;\epsilon) \frac{\partial \mathsf{Z}^{-1}}{\partial g} \mathsf{Z}. \tag{2.30}$$

We will consider the operator of dimension 1

$$\overline{\boldsymbol{\psi}}(x)\boldsymbol{\psi}(x) \tag{2.31}$$

and the operators of dimension 2,

$$K = \mathrm{i}\overline{\boldsymbol{\psi}}(x) \cdot \slashed{\partial}\boldsymbol{\psi}(x), \qquad V = g_0\big(\overline{\boldsymbol{\psi}}(x)\boldsymbol{\psi}(x)\big)^2, \tag{2.32}$$

which appear in the Lagrangian. The renormalization of the fermion bilinear is straightforward, since it is the mass term appearing in the Lagrangian, and we have

$$\overline{\boldsymbol{\psi}}(x)\boldsymbol{\psi}(x) = Z_{\overline{\boldsymbol{\psi}}\boldsymbol{\psi}}[\overline{\boldsymbol{\psi}}(x)\boldsymbol{\psi}(x)] \tag{2.33}$$

where (see e.g. [26])

$$Z_{\overline{\boldsymbol{\psi}}\boldsymbol{\psi}} = Z_m^{-1}. \tag{2.34}$$

Let us now consider the operators $K$ and $V$. They mix under renormalization, and we can calculate the matrix $\mathsf{Z}_{ij}$ very easily by following the method of [27, 28]. In this method one starts with the bare and renormalized effective actions

$$\Gamma^0 = \int \mathrm{d}^d x \Big\{ A \mathrm{i} \overline{\boldsymbol{\psi}} \cdot \slashed{\partial} \boldsymbol{\psi} + B \frac{g_0}{2} \big( \overline{\boldsymbol{\psi}} \cdot \boldsymbol{\psi} \big)^2 + \cdots \Big\},$$

$$\Gamma = \int \mathrm{d}^d x \Big\{ A Z_\psi \mathrm{i} \overline{\boldsymbol{\psi}}_R \cdot \slashed{\partial} \boldsymbol{\psi}_R + B Z_g Z_\psi^2 \frac{g}{2} \big( \overline{\boldsymbol{\psi}}_R \cdot \boldsymbol{\psi}_R \big)^2 + \cdots \Big\}. \tag{2.35}$$

The renormalization constants $Z_\psi$ and $Z_g$ are chosen so that divergences are reabsorbed, and we will fix them in such a way that

$$A Z_\psi = B Z_g Z_\psi^2 = 1. \tag{2.36}$$

Let us consider the generating functional of 1PI Green functions with insertions of the bare operators $K$, $V$ at zero momentum. It can be obtained by acting with appropriate (functional) derivatives with respect to bare quantities on the bare effective action $\Gamma^0$:

$$\Gamma_K^0 = \left[ \frac{1}{2} \int \mathrm{d}^d x \left( \psi_i^\alpha \frac{\delta}{\delta \psi_i^\alpha(x)} + \psi_i^{\dagger\,\alpha} \frac{\delta}{\delta \psi_i^{\dagger\,\alpha}(x)} \right) - 2 g_0 \frac{\partial}{\partial g_0} \right] \Gamma^0,$$

$$\Gamma_V^0 = 2 g_0 \frac{\partial}{\partial g_0} \Gamma^0. \tag{2.37}$$

The renormalized generating functionals for operator insertions of $[K]$ and $[V]$ can be similarly obtained by taking derivatives of $\Gamma$ with respect to renormalized quantities. The renormalization matrix $\mathsf{Z}$ for the Lagrangian operators satisfies

$$\begin{pmatrix} \Gamma_K^0 \\ \Gamma_V^0 \end{pmatrix} = \mathsf{Z} \begin{pmatrix} \Gamma_K \\ \Gamma_V \end{pmatrix}, \tag{2.38}$$

and a simple calculation shows that

$$\mathsf{Z} = \begin{pmatrix} 1 - 2A' & -B' \\ 2A' & 1 + B' \end{pmatrix}, \tag{2.39}$$

where we have denoted

$$A' = g_0 \frac{\partial \log A}{\partial g_0} \tag{2.40}$$

and similarly for $B$. Explicit expressions for these quantities can be obtained from (2.25), (2.26), and one finds

$$A' = \frac{\gamma(g)}{\epsilon}, \qquad B' = \frac{1}{\epsilon} \left( 2\gamma(g) - \frac{\beta(g)}{g} \right). \tag{2.41}$$

We conclude that

$$\begin{pmatrix} K \\ V \end{pmatrix} = \begin{pmatrix} 1 - \frac{2\gamma}{\epsilon} & \frac{1}{\epsilon} \left( \frac{\beta}{g} - 2\gamma \right) \\ \frac{2\gamma}{\epsilon} & 1 + \frac{1}{\epsilon} \left( 2\gamma - \frac{\beta}{g} \right) \end{pmatrix} \begin{pmatrix} [K] \\ [V] \end{pmatrix}, \tag{2.42}$$

and the matrix of anomalous dimensions is given by

$$\gamma = \begin{pmatrix} 2 g \gamma'(g) & 2 g \gamma'(g) - \left( \beta'(g) - \frac{\beta(g)}{g} \right) \\ -2 g \gamma'(g) & -2 g \gamma'(g) + \beta'(g) - \frac{\beta(g)}{g} \end{pmatrix}. \tag{2.43}$$

One can verify explicitly from (2.42) that the operator $K + V$ does not renormalize. This is a consequence of the fact that $K + V$ can be written as the product of an operator times an equation of motion [26]. Indeed, we have

$$\overline{\psi} \frac{\delta S}{\delta \overline{\psi}} = K + V, \tag{2.44}$$

where $S$ is the action.

## 3  Trans-series from the $1/N$ expansion

### 3.1  An exact result for the self-energy

The Gross–Neveu model can be solved exactly at large $N$, and this means that one can calculate correlation functions as a systematic expansion in $1/N$ (see [29, 30] for an excellent presentation). In the large $N$ formulation, one integrates out the fermions in the action (2.4) and writes down the following effective action for the $\sigma$ field:

$$S_{\mathrm{eff}} = -\int \mathrm{d}^2 x \frac{\sigma^2}{2g_0} - \mathrm{i} N \mathrm{Tr} \log(\mathrm{i} S_0^{-1}), \tag{3.1}$$

where

$$S_0(\sigma) = \mathrm{i} \big( \mathrm{i} \slashed{\partial} - \sigma \big)^{-1} \tag{3.2}$$

is the free propagator for a Dirac fermion. This action has two saddle points at large $N$ in which $\sigma$ takes a constant value $\sigma_c = \pm m_0$, and the classical $\mathbb{Z}_2$ symmetry (2.5) is dynamically broken. The value of $m_0$ is determined by the gap equation

$$\frac{1}{N g_0} = \frac{1}{m_0} \int \frac{\mathrm{d}^d k}{(2\pi)^d} \mathrm{Tr} \left[ \frac{\mathrm{i}}{\slashed{k} - m_0} \right]. \tag{3.3}$$

The propagator for the fluctuations of the $\sigma$ field is defined as

$$\Delta^{-1}(x, y) = -\frac{\mathrm{i}}{N} \frac{\delta^2 S_{\mathrm{eff}}}{\delta \sigma(x) \delta \sigma(y)}, \tag{3.4}$$

evaluated at the large $N$ saddle point $\sigma_c = m_0$. In momentum space it is given by

$$\Delta^{-1}(p; m_0) = \frac{\mathrm{i}}{2\pi} \xi \log \left[ \frac{\xi + 1}{\xi - 1} \right], \tag{3.5}$$

where

$$\xi = \sqrt{1 - \frac{4 m_0^2}{p^2}}. \tag{3.6}$$

See Appendix A.1 for some ingredients in the derivation of this formula. The large $N$ theory describes $\sigma$ particles interacting with fermions, the coupling is of order $N^{-1/2}$, and correlation functions can be computed in terms of large $N$ Feynman diagrams. The fermion self-energy has the following form:

$$\Sigma(p) = m_0 + \frac{1}{N} \big( \slashed{p} \Sigma_p + m_0 \Sigma_m \big), \tag{3.7}$$

where $\Sigma_{p,m}$ have a $1/N$ expansion

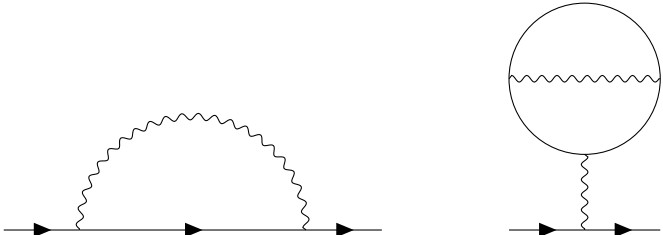

**Figure 1**: Fermion self-energy diagrams at large $N$. The wavy line corresponds to the propagator of the $\sigma$ field, defined in (3.5).

$$\Sigma_{p,m} = \sum_{j \geq 1} \Sigma_{p,m}^j N^{-j+1}. \tag{3.8}$$

The leading order terms can be computed from the diagrams in Fig. 1, and they are given by [17, 29, 31]

$$\Sigma_p^1 = \frac{1}{p^2} \int \frac{\mathrm{d}^2 k}{(2\pi)^2} \frac{p^2 + k \cdot p}{(p+k)^2 - m_0^2} \Delta(k; m_0),$$

$$\Sigma_m^1 = \int \frac{\mathrm{d}^2 k}{(2\pi)^2} \left( \frac{\Delta(k; m_0)}{(p+k)^2 - m_0^2} - \frac{2\pi \mathrm{i}}{k^2 - 4m_0^2} \right). \tag{3.9}$$

These integrals are divergent and they have to be regularized and renormalized. In [17], Campostrini and Rossi found explicit, finite expressions for them by using a sharp momentum cutoff (SM) regularization scheme. In the SM scheme, one first performs a Wick rotation to Euclidean space and computes the angular integral. Then, the resulting integrand is Taylor expanded at infinity. The terms which lead to a divergence are simply subtracted, but in order to avoid IR divergences in the subtracted pieces one has to introduce an IR cutoff $M$ which plays the role of the renormalization scale $\mu$ in dimensional regularization, see [32, 33] for more details. The renormalized self-energy in the SM scheme can be written in terms of the functions

$$A(x) = \frac{1}{4x} \int_0^\infty \left( \xi_y \log \left[ \frac{\xi_y + 1}{\xi_y - 1} \right] \right)^{-1} \left( 1 - \frac{1 + y - x}{\sqrt{(x+y+1)^2 - 4xy}} \right) \mathrm{d}y,$$

$$B(x) = \frac{1}{2} \int_0^\infty \left( \xi_y \log \left[ \frac{\xi_y + 1}{\xi_y - 1} \right] \right)^{-1} \left( \frac{1}{\sqrt{(x+y+1)^2 - 4xy}} + \frac{1 - \xi_y}{2} \right) \mathrm{d}y, \tag{3.10}$$

where

$$\xi_y = \sqrt{1 + \frac{4}{y}}. \tag{3.11}$$

Then the renormalized self-energy has the form,

$$\Sigma^{\mathrm{SM}}(p) = m + \frac{1}{N} \left( \not{p} \Sigma_p^{\mathrm{SM}} + m \Sigma_m^{\mathrm{SM}} \right), \tag{3.12}$$

where $\Sigma_{p,m}^{\mathrm{SM}}$ are given, at leading order in the $1/N$ expansion,

$$\Sigma_p^{\mathrm{SM}} = -A\left( -\frac{p^2}{m^2} \right) + \mathcal{O}(N^{-1}), \qquad \Sigma_m^{\mathrm{SM}} = -B\left( -\frac{p^2}{m^2} \right) + \mathcal{O}(N^{-1}). \tag{3.13}$$

In (3.12), $m$ is the mass gap, which differs from $m_0$ in $1/N$ corrections:

$$m = m_0 + \frac{m_1}{N} + \mathcal{O}(N^{-2}). \tag{3.14}$$

$m_1$ can be calculated in terms of the 't Hooft coupling in the SM scheme, see [17] for details. By relating the SM scheme to the $\overline{\text{MS}}$ scheme, one finds [31, 34]

$$m = \left(1 + \frac{1}{N}\left(\log 2 - \frac{\gamma_E}{2} + \frac{1}{2}\right) + \cdots\right)\Lambda, \tag{3.15}$$

where $\Lambda$ is the dynamically generated scale in the $\overline{\text{MS}}$ scheme, in the conventions of [31, 34]:

$$\Lambda = \mu \left(\frac{\beta_0 g}{2}\right)^{-\beta_1/\beta_0^2} e^{-1/(\beta_0 g)} \exp\left(-\int_0^g \left\{\frac{1}{\beta(x)} + \frac{1}{\beta_0 x^2} - \frac{\beta_1}{\beta_0^2 x}\right\}dx\right), \tag{3.16}$$

and $\beta_0$, $\beta_1$ are the first coefficients of the beta function in (2.12). Let us mention that, in the SM scheme, the beta function and the anomalous dimension of the field are given by [17]

$$\beta_{\text{SM}}(\lambda) = -\lambda^2 + \frac{1}{2N}(2 + \lambda)\lambda^2 + \mathcal{O}(N^{-2}), \qquad \gamma_{\text{SM}}(\lambda) = \mathcal{O}(N^{-2}). \tag{3.17}$$

### 3.2   Trans-series expression

The functions $A(x)$, $B(x)$ give the exact, non-perturbative result for the self-energy of the GN model up to order $1/N$, in the SM scheme. We will now show that these functions can be written as Borel-resummed trans-series, by following the method developed in [1] for the self-energy of the non-linear sigma model (see also [35] for further details on the strategy and results of [1]). For background on trans-series, Borel resummation, and the theory of resurgence, we refer the reader to [36–39].

Let us first focus on $A(x)$. The first step is to write

$$\left[\log\left(\frac{\xi_y + 1}{\xi_y - 1}\right)\right]^{-1} = \int_0^\infty dt \left[\frac{\xi_y - 1}{\xi_y + 1}\right]^t. \tag{3.18}$$

We define the following functions, given as Mellin transforms,

$$K_i(s, t) = \int_0^\infty \xi_y^i \left[\frac{\xi_y - 1}{\xi_y + 1}\right]^t y^{s-1} dy, \qquad i \in \mathbb{Z}, \tag{3.19}$$

which can be computed as quotients of $\Gamma$ functions. We will use

$$K_{-1}(s, t) = \frac{\Gamma(2s + 1)\Gamma(-s + t)}{\Gamma(s + t + 1)}. \tag{3.20}$$

We also define

$$M(s) = \int_0^\infty \frac{y^2}{\sqrt{(x + y + 1)^2 - 4xy}} y^{s-1} dy$$
$$= (1 + x)^{s+1} B(s + 2, -1 - s) {}_2F_1\left(s + 2, -1 - s; 1; \frac{x}{1 + x}\right). \tag{3.21}$$

Then, by using the properties of the Mellin transform and its inverse, we obtain the following representation for the function $A(x)$,

$$A(x) = \frac{1}{4x} \int_0^\infty dt \left[ -\int_{C_1} \frac{ds}{2\pi i} K_{-1}(-s,t)M(s) \right.$$

$$\left. + (x-1) \int_{C_2} \frac{ds}{2\pi i} K_{-1}(-s,t)M(s-1) + K_{-1}(1,t) \right]. \qquad (3.22)$$

In this equation, $C_1$ is the line $c + i\mathbb{R}$ with $c \in (-2,-1)$. For $C_2$, we must consider instead $c \in (-1,0)^3$. We now expand the hypergeometric function in (3.21) at large $x$:

$$M(s) = x^{s+1} \sum_{k \geq 0} a_k(s) x^{-k}, \qquad (3.23)$$

with coefficients

$$a_k(s) = \frac{(-1-s)_k \Gamma(s+2)}{k!^2 \Gamma(s+2-k)} \left[ \log(x) + 2\psi(k+1) - \psi(-s+k-1) - \psi(s+2-k) \right]. \qquad (3.24)$$

We must also consider the combination

$$-M(s) + (x-1)M(s-1) = x^{s+1} \sum_{k \geq 0} \left( -a_k(s) + a_k(s-1) - a_{k-1}(s-1) \right) x^{-k}$$

$$= x^{s+1} \sum_{k \geq 0} b_k(s) x^{-k}, \qquad (3.25)$$

with the convention that $a_{-1}(s) = 0$. Explicitly, the coefficients $b_k(s)$ are given by

$$b_0(s) = 2/(1+s), \qquad (3.26)$$

$$b_k(s) = \frac{2k\Gamma(s+1)\Gamma(k-s-1)}{k!^2 \Gamma(-s)\Gamma(s+1-k)} \left[ \log(x) + 2\psi(k) + \frac{1}{k} - \psi(-s+k-1) - \psi(s-k+1) \right]. \qquad (3.27)$$

Eq. (3.22) is then written as

$$A(x) = \frac{1}{4} \int_0^\infty dt \left[ \sum_{k \geq 0} \int_{C_2} \frac{ds}{2\pi i} K_{-1}(-s,t) b_k(s) x^{-k+s} \right.$$

$$\left. + \int_{C_{-1}} \frac{ds}{2\pi i} K_{-1}(-s,t) a_k(s) x^{-k+s} + \frac{K_{-1}(1,t)}{x} \right], \qquad (3.28)$$

where $C_{-1}$ is a small circle around $s = -1$. We compute the integral in $s$ by using the residue theorem. Deforming the contour $C_2$ to the left, we have contributions from:

---

[3] For $C_1$, the value of $c$ has to be in the region of convergence of the integral $M(s)$ appearing in the first line of (3.21). For $C_2$, we have to consider instead $M(s-1)$, so the region of convergence gets displaced by 1. The values of $c$ will be crucial to determine which singularities have to be included when we deform the contours to the left of the complex plane.

1. Poles of $K_{-1}(-s,t)$ at $s = -t - j$, $j \in \mathbb{Z}_{\geq 0}$. We group the poles contributing with a factor $(-x)^{-n}x^{-t}$, with fixed $n = k + j$ ($0 \leq k \leq n$, $0 \leq j \leq n$). In this way, we define the functions

$$\mathcal{E}_n(t) = \frac{(-1)^n}{4} \sum_{j=0}^{n} b_{n-j}(-t-j)\mathrm{Res}_{s=-t-j}K_{-1}(-s,t). \qquad (3.29)$$

2. Poles of $b_k(s)$ at $s = -j - 1$, $j \in \mathbb{Z}_{\geq 0}$, coming from the last digamma function in (3.27).[4] In this case, we group the poles contributing with a factor $-(-x)^{-n}$, with fixed $n = k + j + 1$ ($0 \leq k \leq n$, $0 \leq j \leq n - 1$). In this way, we define

$$\mathcal{H}_n(t) = -\frac{(-1)^n}{4}\left[ K_{-1}(1,t)\mathrm{Res}_{s=-1}\big(b_{n-1}(s) + a_{n-1}(s)\big)\right.$$

$$\left. + \sum_{j=1}^{n-1} K_{-1}(j+1,t)\mathrm{Res}_{s=-j-1}b_{n-j-1}(s)\right], \qquad n \neq 1, \quad (3.30)$$

with the convention that $a_{-1}(s) = b_{-1}(s) = 0$.

For $n = 1$, we have to add the contribution from the term

$$\tfrac{1}{4}K_{-1}(1,t) = \frac{1}{2t(t^2-1)} \qquad (3.31)$$

in (3.28).

With these conventions, we obtain the final result

$$A(x) = \sum_{n\geq 0}(-x)^{-n} \int_0^{\infty} \mathrm{d}t\big(x^{-t}\mathcal{E}_n(t) - \mathcal{H}_n(t)\big). \qquad (3.32)$$

We now recall that, in order to obtain the self-energy $\Sigma_p^{\mathrm{SM}}$, we have to set $x = -p^2/m^2$. We introduce the variable $\lambda$ as

$$-\frac{m^2}{p^2} = \mathrm{e}^{-2/\lambda} \qquad (3.33)$$

as well as the Borel variable $y = 2t$. Let us note the important fact that, at leading order in the $1/N$ expansion, $\lambda$ can be identified with the running 't Hooft parameter $\lambda(p)$ at the scale set by $p$ in the $\overline{\mathrm{MS}}$ scheme, since at this order one has $m \approx \Lambda$. We also define the functions $E_n(y)$, $F_n(y)$, $G_n(y)$ and $H_n(y)$ as

$$E_n(y) = \frac{\mathcal{E}_n(y/2)}{2} = \frac{1}{\lambda}F_n(y) + G_n(y),$$

$$H_n(y) = \frac{\mathcal{H}_n(y/2)}{2}. \qquad (3.34)$$

We can then write

$$A\left(-\frac{p^2}{m^2}\right) = \sum_{n\geq 0}\left(\frac{m^2}{p^2}\right)^n \int_0^{\infty} \mathrm{d}y\left(\mathrm{e}^{-y/\lambda}E_n(y) - H_n(y)\right). \qquad (3.35)$$

---

[4]Naively, there are more poles coming from the digamma function in $b_k(s)$, but they have residue 0. The case $k = 0$ is an exception in which we only have a single pole at $s = -1$.

One finds the explicit expressions, for $n = 0, 1$,

$$E_0(y) = \frac{1}{2}\frac{1}{2 - y}, \qquad\qquad\qquad\qquad H_0(y) = 0,$$

$$E_1(y) = \frac{1}{\lambda}\frac{y}{4} - \frac{1}{y} - \frac{1}{2} + \frac{y}{8}\left[1 - 2\gamma_E - \psi\left(\frac{y}{2}\right) - \psi\left(-\frac{y}{2}\right)\right], \quad H_1(y) = \frac{4}{y(y-2)(y+2)}, \tag{3.36}$$

where $\psi(x)$ is the digamma function. It is possible to check, on a case by case basis, the equalities

$$\mathrm{Res}_{y=2k}G_n(y) = (-1)^k\mathrm{Res}_{y=2k}H_{n+k}(y), \quad k, n \in \mathbb{Z}_{\geq 0}, \tag{3.37}$$

which are needed to have a complete cancellation of poles in the integral (3.35), after summing over all $n \geq 0$. However, the integrand in (3.35) is singular for a fixed $n$. This allows us to write a formal trans-series out of the above expression, as follows. Let us define

$$r_{n,k} = \mathrm{Res}_{y=2k} H_n(y). \tag{3.38}$$

The first step is to rearrange the integral for fixed $n$ as

$$\int_0^{\infty e^{i\theta}}\left\{e^{-y/\lambda}\left(\frac{F_n(y)}{\lambda} + \widehat{G}_n(y)\right) - \left(H_n(y) - r_{n,0}\frac{e^{-y/\lambda}}{y}\right)\right\}\mathrm{d}y, \tag{3.39}$$

where we have introduced the function

$$\widehat{G}_n(y) = G_n(y) - \frac{r_{n,0}}{y} \tag{3.40}$$

which is regular at the origin. Note that for fixed $n$ the integrand has singularities for positive values of $y$, and we have deformed the integration contour slightly above or below the positive real axis with a small angle $\theta$, to make sense of the integral. The integrand of (3.39)

$$F_n(y) + \lambda\widehat{G}_n(y) = \sum_{k \geq 0}\frac{y^k}{k!}\left(F_n^{(k)}(0) + \lambda\widehat{G}_n^{(k)}(0)\right), \tag{3.41}$$

can be regarded as the Borel transform of the factorially divergent series (our convention for the Borel transform is as in [37])

$$\varphi_n(\lambda) = \sum_{k \geq 0}\left(\lambda^k F_n^{(k)}(0) + \lambda^{k+1}\widehat{G}_n^{(k)}(0)\right) = F_n(0) + \sum_{k \geq 0}\left(F_n^{(k+1)}(0) + \widehat{G}_n^{(k)}(0)\right)\lambda^{k+1}, \tag{3.42}$$

and we can write

$$\int_0^{\infty e^{i\theta}} e^{-y/\lambda}\left(\frac{F_n(y)}{\lambda} + G_n(y) - \frac{r_{n,0}}{y}\right)\mathrm{d}y = s_\pm(\varphi_n)(\lambda), \tag{3.43}$$

where $s_\pm$ are lateral Borel resummations (see e.g. [37] for a definition of these). In this and similar expressions in the following, the $\pm$ sign is correlated with the sign of $\theta$ in the contour deformation.

Let us now consider the second piece, involving $H_n(t)$. A simple calculation shows that

$$-\int_0^{\infty e^{i\theta}}\left(H_n(y) - r_{n,0}\frac{e^{-y/\lambda}}{y}\right)\mathrm{d}y = r_{n,0}\log(\lambda) + c_n \pm i\pi\sum_{k=1}^n r_{n,k}, \tag{3.44}$$

where the constant $c_n$ is defined as the principal value integral

$$c_n = -\mathrm{P} \int_0^\infty \left( H_n(y) - r_{n,0} \frac{\mathrm{e}^{-y}}{y} \right) \mathrm{d}y.$$ (3.45)

It is natural to include the logarithmic and constant pieces in the trans-series, so that the total, factorially divergent series for each $n$ is given by

$$\Phi_{2n}(\lambda) = r_{n,0} \log(\lambda) + c_n \pm \mathrm{i}\pi \sum_{k=1}^n r_{n,k} + \sum_{k \geq 0} \left( \lambda^k F_n^{(k)}(0) + \lambda^{k+1} \widehat{G}_n^{(k)}(0) \right), \qquad n \geq 0.$$ (3.46)

We have labelled these formal series in $\lambda$ with an even index, $2n$, since they multiply even powers of $m$, $m^{2n}$, in the trans-series, and as we will see the function $mB(x)$ leads to odd powers $m^{2n+1}$. For $n = 0, 1$ we use the expressions in (3.36) to obtain

$$\Phi_0(\lambda) = \sum_{k \geq 1} \frac{(k-1)!}{2^{k+1}} \lambda^k,$$

$$\Phi_2(\lambda) = \gamma_E + \log(2) \pm \frac{\mathrm{i}\pi}{2} - \log(\lambda) - \frac{\lambda}{4} + \frac{\lambda^2}{8} + \sum_{k \geq 1} \frac{(2k+1)!}{2^{2k+2}} \zeta(2k+1) \lambda^{2k+2}.$$ (3.47)

The series $\Phi_0(\lambda)$ corresponds to the perturbative sector of the self-energy. It has appeared already in [17], since it gives the classical asymptotic expansion of the function $A(x)$. On the other hand, $\Phi_2(\lambda)$ is the first trans-asymptotic correction to $A(x)$ and multiplies a power correction of order $m^2/p^2$. In addition, it is ambiguous. The ambiguity has a very simple interpretation, in view of the above analysis. The Borel transform of the perturbative series $\Phi_0(\lambda)$ has a singularity at $y = 2$, therefore its two lateral Borel resummations are different. However, the choice of the constant term in the trans-series $\Phi_2(\lambda)$ can be correlated with the choice of lateral resummation of $\Phi_0(\lambda)$, so that the final result is the same for both choices. This is an illustration of the general phenomenon noted by David in [2].

So far we have analyzed the trans-series expression for $A(x)$. In the case of $B(x)$, the trans-series structure can be obtained from the observation that [17]

$$B(x) = \frac{1}{2(1 + 3/x)} \left( 4A(x) + \frac{S(x)}{x} \right),$$ (3.48)

where

$$S(x) = \int_0^\infty \mathrm{d}y \left( \log \left[ \frac{\xi_y + 1}{\xi_y - 1} \right] \right)^{-1} \left[ \frac{y \xi_y}{\sqrt{(x+y+1)^2 - 4xy}} - 1 + \frac{x+1}{2} \left( \frac{1}{\xi_y} - 1 \right) \right].$$ (3.49)

The trans-series structure of $S(x)/x$ is known from [1, 35], and it involves corrections in even powers of $m$. Therefore the trans-series structure for $B(x)$ follows from the results for $A(x)$ and $S(x)/x$. For the purposes of this paper, only the classical asymptotic series of $B(x)$ will be needed. It is given by the formal series in the 't Hooft parameter

$$\Phi_1(\lambda) = \frac{1}{2} - \frac{\gamma_E}{2} - \frac{1}{2} \log(2) + \frac{1}{2} \log(\lambda) + \sum_{k \geq 1} \frac{(2k)!}{2^{2k+1}} \zeta(2k+1) \lambda^{2k+1}.$$ (3.50)

Let us summarize the results in this section. The fermion self-energy $\Sigma(p)$ in the SM scheme, at order $1/N$, is given by an exact expression which is a function of the external momentum $p$ and the mass gap $m$. By using the Mellin transform techniques of [1], this expression can be decoded as a trans-series. This involves a perturbative part, given by the series $\Phi_0(\lambda)$, a power correction proportional to $m \sim \Lambda$ and involving the series $\Phi_1(\lambda)$ in (3.50), and a power correction proportional to $m^2 \sim \Lambda^2$ and involving the series $\Phi_2(\lambda)$ in (3.47). More precisely, we have

$$\Sigma^{\mathrm{SM}}(p) \sim -\frac{1}{N}\not{p}\Phi_0(\lambda) + m\left(1 - \frac{1}{N}\Phi_1(\lambda)\right) - \frac{1}{N}\not{p}\frac{m^2}{p^2}\Phi_2(\lambda) + \cdots. \tag{3.51}$$

We note that the l.h.s. of this equation is a well-defined function of $p$, while on the r.h.s. we have a trans-series representation. The first term is the perturbative series, while the second and the third term are non-perturbative power corrections. The dots refer to higher order power corrections, and to higher order corrections in $1/N$. Thanks to the exact large $N$ analysis, we have precise, all-loop predictions for the power series attached to each of these power corrections.

We should be able to reproduce the first term in the r.h.s. of (3.51) from a conventional perturbative calculation in, say, the $\overline{\mathrm{MS}}$ scheme. This was essentially verified in [17], although we will present a more detailed calculation in the next section. In addition, if the method of OPE with vacuum condensates proposed in [10–12] gives the correct trans-series representation of observables, the power corrections in the r.h.s. of (3.51) should be calculable with the SVZ approach, up to unknown overall constants related to the values of the vacuum condensates. We will also verify this in the next section.

# 4   Trans-series from condensates

In this section we will calculate the series $\Phi_{0,1,2}(\lambda)$ in perturbation theory with condensates. We will always work in the $\overline{\mathrm{MS}}$ scheme.

## 4.1   Perturbative series

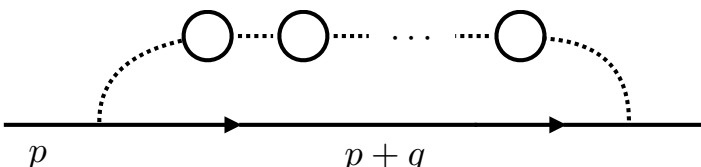

**Figure 2**: The $1/N$ correction to the fermion self-energy is given by a chain of $n$ bubbles.

The first step is to compute the leading $1/N$ correction to the fermion self-energy, at all orders in the 't Hooft coupling constant. This is a standard renormalon calculation, since the $n$-th order correction is due to a chain of $n$ fermion loops or "bubbles," linked by $\sigma$ propagators, as shown in Fig. 2. Using the Feynman rules, one finds that the contribution of $n$ loops is

$$(\mathrm{i}\sqrt{g_0})^{2n+2}(-\mathrm{i})^{n+1}N^n\Pi^n(q^2) = \frac{\mathrm{i}}{N}\left(\mathrm{i}\Pi(q^2)\right)^n(\pi\lambda_0)^{n+1}, \tag{4.1}$$

where we have introduced the bare 't Hooft parameter $\lambda_0$ as in (2.14), and $\Pi(q^2)$ is the fermion polarization loop (A.5). We now express $\lambda_0$ in terms of the renormalized coupling constant through (2.27), and we sum the geometric series of bubbles to

$$\frac{\mathrm{i}}{N}\frac{1}{(\pi\lambda(\nu^2)^{\epsilon/2})^{-1}Z_\lambda^{-1} - \mathrm{i}\Pi(q^2)}. \tag{4.2}$$

If we now take into account that

$$-\mathrm{i}\Pi(q^2) = -\frac{1}{\pi\epsilon} + \cdots, \tag{4.3}$$

we deduce that the renormalization constant $Z_\lambda$ is given, at large $N$, by

$$Z_\lambda^{-1} = 1 + \frac{\lambda}{\epsilon} + \cdots, \tag{4.4}$$

in agreement with the result (2.17) for the beta function at leading order in the $1/N$ expansion.

We will write the (bare) self-energy as in (3.7),

$$\Sigma(p) \sim \frac{1}{N}\slashed{p}\Sigma_p, \tag{4.5}$$

where the asymptotic sign $\sim$ emphasizes that our calculation will lead to a representation in terms of formal power series. If we define

$$\mathcal{I}_0(n) = \frac{1}{\slashed{p}}\pi^n \int \frac{\mathrm{d}^d q}{(2\pi)^d} \frac{(\slashed{p}+\slashed{q})}{(p+q)^2}(\mathrm{i}\Pi(q^2))^{n-1}, \tag{4.6}$$

it is easy to see that the leading term in the $1/N$ expansion of $\Sigma_p$ is

$$\Sigma_p^{\mathrm{P},1} = \sum_{n \geq 2} \mathrm{i}\mathcal{I}_0(n)\lambda_0^n, \tag{4.7}$$

where we have added a superscript P to indicate that this is the perturbative result, and the superindex 1 means that, as in (3.8), this is the term of order $1/N$ in the $1/N$ expansion. We can perform the integral (4.6) explicitly by using (A.3), and we find

$$\mathcal{I}_0(n) = \frac{\mathrm{i}}{4}\left(-\frac{p^2}{4\pi}\right)^{-n\epsilon/2}\frac{\Gamma\left(1-\frac{\epsilon}{2}\right)\Gamma\left(\frac{n\epsilon}{2}\right)\Gamma\left(1-\frac{n\epsilon}{2}\right)}{\Gamma\left(\frac{(n-1)\epsilon}{2}\right)\Gamma\left(2-\frac{(n+1)\epsilon}{2}\right)}\left[-\frac{1}{2}\frac{\Gamma\left(1-\frac{\epsilon}{2}\right)\Gamma\left(1+\frac{\epsilon}{2}\right)\Gamma\left(-\frac{\epsilon}{2}\right)}{\Gamma(1-\epsilon)}\right]^{n-1}. \tag{4.8}$$

We can now use the techniques of Appendix A.2 to write the sum (4.7) in terms of the structure function

$$F(x,y) = -\frac{1}{2}\left(-\frac{p^2}{4\pi\nu^2}\right)^{-y/2}\frac{\Gamma\left(1-\frac{x}{2}\right)\Gamma\left(1+\frac{y}{2}\right)\Gamma\left(1-\frac{y}{2}\right)}{\Gamma\left(\frac{y-x}{2}\right)\Gamma\left(2-\frac{y+x}{2}\right)}\left[\frac{\Gamma\left(1+\frac{x}{2}\right)\Gamma^2\left(1-\frac{x}{2}\right)}{\Gamma(1-x)}\right]^{y/x-1}. \tag{4.9}$$

This function contains both the divergent and the finite part of the bare self-energy. To renormalize the self-energy we need to introduce the renormalization of the field, which has the $1/N$ expansion

$$Z_\psi = 1 + \frac{1}{N}Z_\psi^{(1)} + \cdots. \tag{4.10}$$

The renormalized self-energy is then given by

$$\Sigma_{p,R}^{\mathrm{P},1} = \Sigma_p^{\mathrm{P},1} - Z_\psi^{(1)}. \tag{4.11}$$

By using the results of Appendix A.2 we find

$$Z_\psi^{(1)} = \left[\Sigma_p^{\mathrm{P},1}\right]_{\mathrm{div}} = \left[F_0(\epsilon)\log\left(1+\frac{\lambda}{\epsilon}\right)\right]_{\mathrm{div}}, \tag{4.12}$$

where the function $F_0(\epsilon) = F(\epsilon, 0)$ is given by

$$F_0(\epsilon) = \frac{\epsilon}{2} \frac{1}{2-\epsilon} \frac{\Gamma(1-\epsilon)}{\Gamma^3(1-\frac{\epsilon}{2})\Gamma(1+\frac{\epsilon}{2})} \tag{4.13}$$

and we have taken into account that $F_{0,0} = F(0,0) = 0$. We can now use (A.24) to obtain the anomalous dimension of the field, at the first non-trivial order in $1/N$, as

$$\gamma^{(1)}(\lambda) = -\lambda F_0(-\lambda) \tag{4.14}$$

which reproduces the result (2.24).

Let us now compute the renormalized self-energy, given by the finite part as $\epsilon \to 0$. The finite part can also be computed in terms of the structure function by using the general formula (A.23). We first note that, from (4.14) and $\beta_\lambda^{(0)}(\lambda) = -\lambda^2$, we have

$$-\int_0^\lambda \frac{F_0(-u)}{u} du = -\int_0^\lambda \frac{\gamma^{(1)}(u)}{\beta_\lambda^{(0)}(u)} du. \tag{4.15}$$

Therefore, in this case, (A.23) reads

$$\Sigma_{p,R}^{\mathrm{P},1} = -\int_0^\lambda \frac{\gamma^{(1)}(u)}{\beta_\lambda^{(0)}(u)} du - F_{0,1}\lambda + \sum_{m\geq 2}(m-1)! F_m(0)\lambda^m. \tag{4.16}$$

From the structure function (4.9), we find

$$-F_{0,1}\lambda = -\frac{\lambda}{4}, \qquad \sum_{m\geq 2} F_m(0)y^m = -\frac{y}{2(2-y)} + \frac{y}{4}, \tag{4.17}$$

where we have set $p^2 = -\mu^2$ in (4.9) and therefore the coupling constant appearing above is $\lambda = \lambda(p)$. Since $F_{0,1} = -F_{1,0}$, we obtain

$$\Sigma_{p,R}^{\mathrm{P},1} = -\int_0^{\lambda(p)} \frac{\gamma^{(1)}(u)}{\beta_\lambda^{(0)}(u)} du - \sum_{m\geq 1} \frac{(m-1)!}{2^{m+1}}(\lambda(p))^m. \tag{4.18}$$

To compare this result with (3.51) we have to take into account the change of scheme, from SM to $\overline{\mathrm{MS}}$. The coupling constants $\lambda$ appearing in both expressions are the same, up to this order in $1/N$. In a change of scheme characterized by a coupling $g$ to a new one characterized by $g'$, $\ell$-point Green functions pick a factor $\zeta^\ell(g)$, where $\zeta(g) = 1 + \mathcal{O}(g)$ is determined by the equation (see e.g. [26])

$$\gamma'(g') - \gamma(g) = -2\beta(g)\frac{\partial}{\partial g}\log\zeta(g), \tag{4.19}$$

and $\gamma'(g')$ is the anomalous dimension in the new scheme. In our case, using the SM anomalous dimension in (3.17), we find that

$$\zeta(\lambda) = 1 + \frac{1}{2N}\int_0^\lambda \frac{\gamma^{(1)}(u)}{\beta_\lambda^{(0)}(u)} du + \mathcal{O}(N^{-2}). \tag{4.20}$$

Therefore, the relation between the self-energy $\Sigma_p^{\text{SM}}$ in the SM scheme and the self-energy $\Sigma_{p,R}$ in the $\overline{\text{MS}}$ scheme, at leading order in the $1/N$ expansion, is given by

$$\Sigma_p^{\text{SM}} = \Sigma_{p,R} + \int_0^\lambda \frac{\gamma^{(1)}(u)}{\beta_\lambda^{(0)}(u)} \mathrm{d}u + \mathcal{O}(N^{-1}).\qquad(4.21)$$

Since the power series in the second term of the r.h.s. of (4.18) is nothing but $-\Phi_0(\lambda)$, we find that the perturbative calculation of the self-energy in the $\overline{\text{MS}}$ scheme gives precisely the perturbative part of the trans-series (3.51), after taking into account the correction (4.21).

Let us mention that the factorially divergent perturbative series (4.18) is perhaps the simplest example of an IR renormalon in an asymptotically free theory. As is well-known, IR renormalons are smoking guns for non-perturbative corrections due to condensates. We will see later on in this paper that in the case of the renormalon (4.18), the corresponding condensate is the four-quark condensate[5].

## 4.2 General aspects of perturbation theory with condensates

We would like now to calculate the power corrections appearing in (3.51) within the SVZ approach. The basic idea in this approach is to use the OPE for the operators appearing in the two-point function. In our case the OPE reads, schematically,

$$\overline{\psi}(x)\psi(0) = C(x)\mathbf{1} + C_{\overline{\psi}\psi}(x)[\overline{\psi}(0)\psi(0)] + C_K(x)[K(0)] + C_V(x)[V(0)] + \cdots,\qquad(4.22)$$

where we have included the operators of dimension one and two. One further assumes that the different operators appearing in the OPE have non-vanishing vevs, also called vacuum condensates. In the GN model this is expected to be so, as it can seen for example in the large $N$ solution of the model. Indeed, it is elementary to show that, in the $\overline{\text{MS}}$ scheme, the solution of the gap equation (3.3) is

$$\sigma_c^2 = \Lambda^2.\qquad(4.23)$$

On the other hand, $\sigma$ can be integrated out in (2.4) and is given by

$$\sigma = \sqrt{g_0}\overline{\psi}\psi,\qquad(4.24)$$

therefore we have

$$\langle[\overline{\psi}\psi]\rangle_c \approx -\frac{N}{\pi\lambda}\Lambda\qquad(4.25)$$

at large $N$. The operator $[V]$ appearing in the Lagrangian should also have a non-trivial vev, by large $N$ factorization, and we expect

$$\langle[V]\rangle_c \approx \frac{\pi\lambda}{N}\langle[\overline{\psi}\psi]\rangle_c^2\qquad(4.26)$$

at large $N$. In addition, due to (2.44), we also expect

$$\langle[K]\rangle_c = -\langle[V]\rangle_c.\qquad(4.27)$$

---

[5]In section IV of the original paper by Gross and Neveu [16] they consider the self-energy for the theory in which fermions have a mass term of the form (2.6). They show that the perturbative expansion of this quantity is factorially divergent, and this is the first appearance of a renormalon in the QFT literature. However their example is not an IR renormalon, but an UV renormalon.

Therefore, we will assume that all the operators appearing in the OPE (4.22) lead to non-trivial vacuum condensates, and we will assume that these condensates satisfy the properties (4.26), (4.27), as they follow from large $N$ factorization and basic principles.

In order to make contact with the trans-series (3.51) we need the precise relation between vacuum condensates and the dynamically generated scale $\Lambda$, which can be in turn related to the mass gap through (3.15). This relation follows from general principles, since the vevs of composite operators have to satisfy the Callan–Symanzik equation

$$\left[\delta_{ij}\left(\mu\frac{\partial}{\partial\mu} + \beta(g)\frac{\partial}{\partial g}\right) + \gamma_{ij}\right]\langle[\mathcal{O}_j]\rangle_c = 0, \tag{4.28}$$

in the general case of operator mixing. If there is a single operator $\mathcal{O}$ of dimension $d$, with anomalous dimension $\gamma_{\mathcal{O}}$, the solution of the equation (4.28) is

$$\langle[\mathcal{O}]\rangle_c = \xi\Lambda^d \exp\left(-\int^{g(\mu)} \frac{\gamma_{\mathcal{O}}(u)}{\beta(u)}\mathrm{d}u\right), \tag{4.29}$$

where $\xi$ is an overall constant which might depend on the parameters of the theory, like $N$. Eq. (4.29) applies in particular to the operator $\overline{\psi}\psi$, which does not mix. In this case we have $\gamma_{\overline{\psi}\psi} = -\gamma_m$. By using the explicit expressions (2.20), (2.21) for the mass anomalous dimension, one finds

$$\langle[\overline{\psi}\psi]\rangle_c = -Nc(N)\frac{\Lambda}{\pi\lambda} \exp\left(-\frac{1}{N}\int^\lambda \frac{\chi(u)}{u^2}\mathrm{d}u + \mathcal{O}(N^{-2})\right), \tag{4.30}$$

where $\chi(u)$ is given in (2.22). Its expansion around $u = 0$ has the form

$$\chi(u) = \chi_1 u + \mathcal{O}(u^2), \qquad \chi_1 = \frac{1}{2}, \tag{4.31}$$

and the integral appearing in (4.30) has to be understood as

$$\int^\lambda \frac{\chi(u)}{u^2}\mathrm{d}u = \chi_1 \log(\lambda) + \int_0^\lambda \frac{\chi(u) - \chi_1 u}{u^2}\mathrm{d}u. \tag{4.32}$$

If we compare (4.30) with the result at large $N$ in (4.25) we find that $c(N)$ has the $1/N$ expansion

$$c(N) = 1 + \frac{c_1}{N} + \mathcal{O}(N^{-2}). \tag{4.33}$$

The choice of normalization in (4.30), with additional factors of $\pi$, $N$ was made so that $c(N) \approx 1$ at large $N$, as shown in (4.33). The subleading terms in the expansion of $c(N)$ could be obtained by calculating $1/N$ corrections to the effective potential.

In the case of the operators $K$, $V$, there is operator mixing and the matrix of anomalous dimensions is given in (2.43). We can reduce the system of equations to a single equation by taking into account (4.27), and solve it in the $1/N$ expansion. A simple calculation gives

$$\langle[V]\rangle_c = Nd(N)\frac{\Lambda^2}{\pi\lambda} \exp\left(\frac{1}{N}\left(\frac{\beta^{(1)}(\lambda)}{\lambda^2} - 1\right) + \mathcal{O}(N^{-2})\right), \tag{4.34}$$

where

$$d(N) = 1 + \frac{d_1}{N} + \mathcal{O}(N^{-2}). \tag{4.35}$$

The fact that $d(N) \approx 1$ at large $N$ is a consequence of the large $N$ factorization (4.26).

With the above results, we can calculate the value of the condensates appearing in the OPE (4.22), up to the non-perturbative functions $c(N)$, $d(N)$. The Wilson coefficients can be determined in various ways. The most practical method is the following (see [40] for an excellent exposition in the context of QCD): one expands in series the interaction terms in the Lagrangian, as in standard perturbation theory. We then Wick-contract the elementary fields with the usual rules, except for the fields that will enter into the condensate. For example, to calculate $C_{\overline{\psi}\psi}$, corresponding to the two-quark condensate, two of the elementary fields shouldn't participate in the contraction, but form the vacuum condensate. The combinatorial possibilities for doing this expansion can be represented by Feynman diagrams in which the fields that form the condensate are represented by blobs. We will see plenty of examples of this procedure in the next section.

In forming the condensates, we will encounter vevs of fermion operators which are not Lorentz scalars or $U(N)$ singlets. These vevs can be determined easily by imposing Lorentz and $U(N)$ invariance. For example, for a general bilinear in fermions, we have

$$\langle \psi_i^\mu(0)\overline{\psi}_j^\nu(0)\rangle_c = -\frac{\delta_{ij}\delta^{\mu\nu}}{2N}\langle\overline{\psi}\psi\rangle_c, \tag{4.36}$$

while for the operators related to the Lagrangian operators $V$, $K$ we have

$$g_0\langle \psi_i^\alpha(0)\overline{\psi}_j^\beta(0)\psi_k^\mu(0)\overline{\psi}_l^\nu(0)\rangle_c = \left(\delta^{\alpha\beta}\delta^{\mu\nu}\delta_{ij}\delta_{kl} - \delta^{\alpha\nu}\delta^{\mu\beta}\delta_{il}\delta_{jk}\right)\frac{\langle V\rangle_c}{2N(2N-1)},$$

$$\langle \partial_\mu\psi_i^\alpha(0)\overline{\psi}_j^\beta(0)\rangle_c = \frac{\mathrm{i}}{2dN}(\gamma_\mu)^{\alpha\beta}\delta_{ij}\langle K\rangle_c. \tag{4.37}$$

We note that we use a dimensional regularization in which the dimension of space-time is $d$, but the dimension of the Dirac spinor space is $2^{\lfloor d/2 \rfloor} = 2$. To relate the vevs of the bare operators to the finite vevs of the renormalized operators we use the renormalization constants in (2.33), (2.42). For the operators $K$, $V$, using (2.44), we find

$$\langle V\rangle_c = -\langle K\rangle_c = \left(1 - \frac{\beta_\lambda(\lambda)}{\epsilon\lambda}\right)\langle[V]\rangle_c. \tag{4.38}$$

## 4.3   Trans-series for the two-quark condensate

Let us now calculate the contribution to the self-energy of the trans-series associated to the two-quark condensate $\langle\overline{\psi}\psi\rangle_c$. Similar calculations in QCD can be found in [10, 41, 42]. To illustrate the OPE method with condensates, we will first consider in some detail the contribution at leading order in the coupling constant. A pedagogical exposition of the method in the case of QCD can be found in [40].

It is convenient to start the calculation in position space, and only later Fourier transform into momentum space. We recall the Wick contractions are defined as

$$\overbrace{\sigma(x)\sigma}(y) = -\mathrm{i}\delta(x-y) \tag{4.39}$$

and

$$\overbrace{\psi_i^\mu(x)\overline{\psi}_j^\nu}(y) = \delta_{ij}S_0^{\mu\nu}(x-y), \tag{4.40}$$

where

$$S_0^{\mu\nu}(x) = \int \frac{\mathrm{d}^d k}{(2\pi)^d}\mathrm{e}^{-\mathrm{i}kx}\left[\frac{\mathrm{i}}{\not{k}}\right]^{\mu\nu}. \tag{4.41}$$

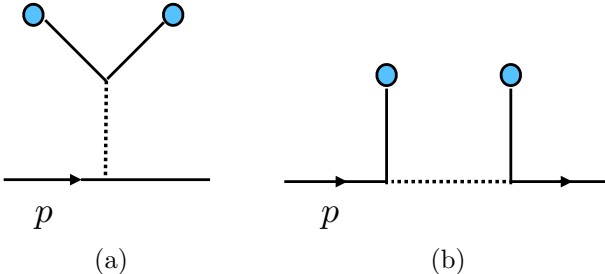

<p style="text-align:center">(a)          (b)</p>

**Figure 3**: The two diagrams that contribute to the two-quark condensate correction to the two-point function, at leading order in $g_0$.

To compute the two-point function at order $g_0$, we bring down two factors of the interaction term in the Lagrangian. This yields the following expression

$$-g_0 \langle \psi_i^\mu(x) \overline{\psi}_j^\nu(0) \int \mathrm{d}^d y_1 \mathrm{d}^d y_2 \, \overline{\psi}_m^\alpha(y_1) \psi_m^\alpha(y_1) \sigma(y_1) \overline{\psi}_n^\beta(y_2) \psi_n^\beta(y_2) \sigma(y_2) \rangle, \qquad (4.42)$$

where we have to perform Wick contractions, but leaving a two-quark pair uncontracted to form the condensate. There are two ways of doing this. In the first way, we contract both external legs to the same vertex. One set of contractions is

$$\langle \psi_i^\mu(x) \overline{\psi}_j^\nu(0) \overline{\psi}_m^\alpha(y_1) \psi_m^\alpha(y_1) \sigma(y_1) \overline{\psi}_b^\nu(y_2) \psi_b^\nu(y_2) \sigma(y_2) \rangle \qquad (4.43)$$

and there is another, equivalent one obtained by exchange of the vertices $y_1$ and $y_2$. This type of contractions can be represented by the diagram in Fig. 3a. Note that the two-quark pair which leads to the vacuum condensate is traced over. After performing a Fourier transform into momentum space and doing the integrals in the spacetime variables, we obtain

$$-\frac{\mathrm{i} g_0}{p^2} \langle \overline{\boldsymbol{\psi}} \boldsymbol{\psi} \rangle_c. \qquad (4.44)$$

This is a contribution of order one in the $1/N$ expansion and, after renormalization, it will give the term $m$ in the trans-series of the self-energy (3.51).

In the second type of contributions, we contract each external leg with a different vertex. This leads to contractions of the form

$$\langle \psi_i^\mu(x) \overline{\psi}_j^\nu(0) \overline{\psi}_m^\alpha(y_1) \psi_m^\alpha(y_1) \sigma(y_1) \overline{\psi}_b^\beta(y_2) \psi_b^\beta(y_2) \sigma(y_2) \rangle \qquad (4.45)$$

as well as a similar one obtained by exchanging the vertices. They can be represented by the diagram in Fig. 3b. The vev of the product $\psi_m^\alpha(y_1) \overline{\psi}_b^\beta(y_2)$ leads to a condensate and, due to the delta function $\delta(y_1 - y_2)$ coming from (4.39), both fields are at the same point. After using (4.36), we find that the contribution of the diagram in Fig. 3b to the two-point function is simply

$$\frac{\mathrm{i} g_0}{2N} \frac{1}{p^2} \langle \overline{\boldsymbol{\psi}} \boldsymbol{\psi} \rangle_c. \qquad (4.46)$$

A general principle to retain from this calculation is that condensates in which both quarks come from the same interaction vertex have a relative factor of $N$, as compared to condensates where

the fermions are from different vertices. This is important when taking into account large $N$ counting.

After multiplying the diagrams in Fig. 3 by i and removing their external legs, we find that the total contribution of the two-quark trans-series to the self-energy, at first order in the coupling, is

$$-g_0\left(1 - \frac{1}{2N}\right)\langle\overline{\psi}\psi\rangle_c. \tag{4.47}$$

Let us mention that the diagram in Fig. 3b has a counterpart in the calculation of the two-quark condensate correction to the quark propagator in QCD [10, 41], in which the $\sigma$ propagator is replaced with a gluon.

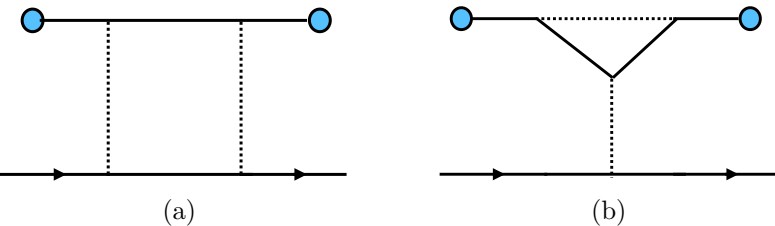

(a)            (b)

**Figure 4**: These diagrams are of order $1/N$, but they do not contribute to the two-quark condensate trans-series. Diagram (a) vanishes since it is proportional to $\mathrm{Tr}\,\gamma^{\mu} = 0$, and (b) involves propagators at zero momentum.

After this pedagogical exercise, let us consider the diagrams which give the trans-series $\Phi_1(\lambda)$ in the self-energy (3.51). In order to proceed, we recall that in calculating the self-energy in conventional perturbation theory, one considers only one-particle irreducible diagrams, i.e. diagrams that can not be divided into subdiagrams by cutting one fermionic internal line. The underlying reason is that reducible diagrams factorize, therefore their contribution can be obtained from their subdiagrams. In the presence of condensates one has to be careful, since diagrams that look reducible in the conventional sense do not factorize. Let us then reconsider the relationship between the two-point function and the self-energy when one includes non-perturbative corrections coming from condensates.

As it is clear from (3.51), the self-energy up to order $1/N$ is a trans-series. The inverse, renormalized two-point function can then be written as

$$\mathrm{i}S_R^{-1}(p) = \not{p} - \frac{1}{N}\not{p}\Sigma_{p,R}^{\mathrm{P}} - \mathcal{C}\Sigma_{m,R}^{\mathrm{NP}} - \frac{\mathcal{C}^2}{N}\not{p}\Sigma_{p,R}^{\mathrm{NP}} + \mathcal{O}(\mathcal{C}^3). \tag{4.48}$$

In this equation, we have introduced a trans-series parameter $\mathcal{C}$ to keep track of the powers of $m$, the superscripts P, NP refer to perturbative and non-perturbative contributions, respectively, and the subscript $R$ stands for renormalized. The self-energies appearing in this equation have $1/N$ expansions with the structure

$$\begin{aligned}
\Sigma_{p,R}^{\mathrm{P}} &= \sum_{j\geq 1}\Sigma_{p,R}^{\mathrm{P},j}N^{-j+1}, \\
\Sigma_{m,R}^{\mathrm{NP}} &= \sum_{j\geq 0}\Sigma_{m,R}^{\mathrm{NP},j}N^{-j}, \\
\Sigma_{p,R}^{\mathrm{NP}} &= \sum_{j\geq 1}\Sigma_{p,R}^{\mathrm{NP},j}N^{-j+1}.
\end{aligned} \tag{4.49}$$

An important remark is that, in doing the $1/N$ expansions of the self-energies computed diagrammatically, we keep the condensates themselves fixed, i.e. we do not expand them as in (4.30) or (4.34). This is the natural $1/N$ counting when working with diagrams with condensates. We now expand $S_R(p)$ in $\mathcal{C}$ and $1/N$ to obtain

$$S_R(p) = \frac{\mathrm{i}}{\not{p}}\left\{1 + \frac{1}{N}\Sigma_{p,R}^{\mathrm{P},1} + \frac{\mathcal{C}}{\not{p}}\left(\Sigma_{m,R}^{\mathrm{NP},0} + \frac{1}{N}\Sigma_{m,R}^{\mathrm{NP},1}\right) + \frac{2\mathcal{C}}{N\not{p}}\Sigma_{m,R}^{\mathrm{NP},0}\Sigma_{p,R}^{\mathrm{P},1} + \mathcal{O}\!\left(\mathcal{C}^2, N^{-2}\right)\right\}. \qquad (4.50)$$

The second term inside the brackets is the perturbative piece calculated in (4.18), and $\Sigma_{m,R}^{\mathrm{NP},0}$ can be obtained from (4.44). The fourth term appearing inside the brackets in the r.h.s. factorizes into a perturbative piece and a non-perturbative piece, and as we will see it corresponds to a reducible diagram.

Let us now consider the diagrams that contribute to (4.50). They have to be of order $1/N$, but incorporate all loops. In conventional perturbation theory, such diagrams are obtained by inserting chain bubbles, and the same principle holds in the case of perturbation theory with condensates. There are however various diagrams that have the right $1/N$ counting but vanish in dimensional regularization, or vanish because they involve a trace of an odd number of gamma matrices. An example is the diagram in Fig. 4a. There is another type of diagrams that do not contribute: condensates are essentially zero-momentum insertions and they can lead to diagrams in which we have propagators at zero momentum. These diagrams have to be discarded [40]. The diagram in Fig. 4b is an example of this. We note however that the diagram in Fig. 4a turns out to contribute to the four-quark condensate correction, as we will explain in section 4.4.

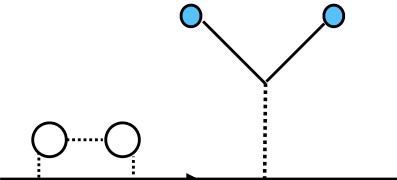

**Figure 5**: A reducible diagram.

Among the non-vanishing diagrams, one finds Fig. 5. Its contribution factorizes into the contribution of the perturbative diagram of Fig. 2, which is of order $1/N$, and the contribution of the diagram in Fig. 3a. This is the reducible diagram that corresponds to the fourth term inside the bracket in (4.50), as we anticipated above. It does not contribute to the self-energy.

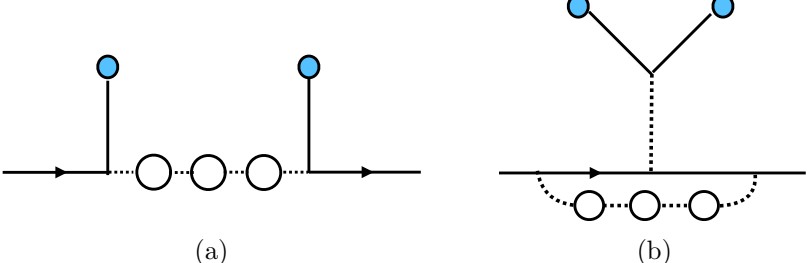

**Figure 6**: Irreducible diagrams that contribute to the two-quark condensate trans-series at order $1/N$, and all loops.

Let us then consider diagrams which do not factorize. We will call such diagrams irreducible. They are shown in Fig. 6 (in the drawings we show only insertions of two or three bubbles, but of

course one should consider insertions of an arbitrary number of bubbles). The diagram in Fig. 6a is obtained by inserting the bubble chain (4.1) inside in Fig. 3b, and no additional integration is needed. Its contribution to the self-energy is

$$\frac{\pi\lambda_0}{2N^2}\langle\overline{\boldsymbol{\psi}}\boldsymbol{\psi}\rangle_c \sum_{n\geq 1}\big(\mathrm{i}\Pi(p^2)\big)^n(\pi\lambda_0)^n, \tag{4.51}$$

where $n$ is the number of polarization loops inserted.[6] Notice that, when $n \geq 1$, the fermion and antifermion fields in the condensate are no longer at the same point. In this case, we have to expand

$$\langle\psi_m^\alpha(y_1)\overline{\psi}_n^\beta(y_2)\rangle_c = \langle\psi_m^\alpha(0)\overline{\psi}_n^\beta(0)\rangle_c + (y_1-y_2)^\mu\langle\partial_\mu\psi_m^\alpha(0)\overline{\psi}_n^\beta(0)\rangle_c + \cdots$$
$$= -\frac{1}{2N}\delta_{mn}\delta^{\alpha\beta}\langle\overline{\boldsymbol{\psi}}\boldsymbol{\psi}\rangle_c + \cdots . \tag{4.52}$$

In calculating the contribution to the two-quark condensate we only retain the first term in the r.h.s. of the first line, but the derivative term will contribute to the four-quark condensate, as we will see in the next section.

Let us now consider the diagram in Fig. 6b. A straightforward calculation gives the following contribution to the self-energy:

$$\frac{\pi\lambda_0}{N^2}\langle\overline{\boldsymbol{\psi}}\boldsymbol{\psi}\rangle_c \sum_{n\geq 1}\mathrm{i}\mathcal{I}_1(n)\lambda_0^n, \tag{4.53}$$

where $\mathcal{I}_1(n)$ is the loop integral

$$\mathcal{I}_1(n) = \pi^n \int \frac{\mathrm{d}^d q}{(2\pi)^d}\frac{\big(\mathrm{i}\Pi(q^2)\big)^{n-1}}{(p+q)^2}. \tag{4.54}$$

This integral can be evaluated in dimensional regularization using (A.3):

$$\mathcal{I}_1(n) = -\frac{\mathrm{i}}{4}\left(-\frac{p^2}{4\pi}\right)^{-\epsilon n/2}\frac{\Gamma\big(-\frac{\epsilon}{2}\big)\Gamma\big(\frac{n\epsilon}{2}\big)\Gamma\big(1-\frac{n\epsilon}{2}\big)}{\Gamma\big(\frac{(n-1)\epsilon}{2}\big)\Gamma\big(1-\frac{(n+1)\epsilon}{2}\big)}\left[-\frac{1}{2}\frac{\Gamma\big(1-\frac{\epsilon}{2}\big)\Gamma\big(1+\frac{\epsilon}{2}\big)\Gamma\big(-\frac{\epsilon}{2}\big)}{\Gamma(1-\epsilon)}\right]^{n-1}. \tag{4.55}$$

We can now add the contributions of the two diagrams, together with (4.47), to obtain the bare, non-perturbative correction to the term $\Sigma_m$ in the self-energy:

$$\Sigma_m^{\mathrm{NP}} = -\frac{\pi\lambda_0}{N}\langle\overline{\boldsymbol{\psi}}\boldsymbol{\psi}\rangle_c\left\{1 - \frac{1}{2N} - \frac{1}{N}\sum_{n\geq 1}\left(\mathrm{i}\mathcal{I}_1(n) + \frac{1}{2}(\mathrm{i}\pi\Pi)^n\right)\lambda_0^n + \mathcal{O}\big(N^{-2}\big)\right\}. \tag{4.56}$$

There are three sources of renormalization in this quantity: renormalization of the coupling constant, renormalization of the self-energy, and renormalization of the composite operator appearing in the condensate (see [43] for useful remarks on renormalization of fermion propagators in perturbation theory with condensates). The renormalization of the self-energy is done as in the perturbative case, and it just follows by multiplying the inverse two-point function by $Z_\psi$. The renormalization of the composite operator follows from (2.34):

$$\langle\overline{\boldsymbol{\psi}}\boldsymbol{\psi}\rangle_c = Z_m^{-1}\langle[\overline{\boldsymbol{\psi}}\boldsymbol{\psi}]\rangle_c. \tag{4.57}$$

---

[6] In this equation, we start the sum at $n = 1$, since the term $n = 0$ will be already accounted for when we add (4.47).

The renormalized result is

$$\Sigma_{m,R}^{\mathrm{NP}} = -\frac{\pi\lambda}{N}\langle[\overline{\psi}\psi]\rangle_c\frac{Z_\lambda}{Z_m}\left\{1 - \frac{1}{2N} + \frac{1}{N}Z_\psi^{(1)} - \frac{1}{N}\sum_{n\geq 1}\left(\mathrm{i}\mathcal{I}_1(n) + \frac{1}{2}(\mathrm{i}\pi\Pi)^n\right)\lambda_0^n + \mathcal{O}(N^{-2})\right\}. \quad (4.58)$$

Let us note that the sign of the renormalization constant $Z_\psi^{(1)}$ is the opposite one to what is found for the perturbative part in (4.11). This can be seen by comparing (4.50) to the diagrammatic expansion, or simply by noting that $\not{p}$ and $m$ have opposite signs in the inverse propagator. The renormalization constants $Z_\lambda$, $Z_m$ have a $1/N$ expansion

$$Z_\lambda = Z_\lambda^{(0)}\left(1 + \frac{1}{N}\widehat{Z}_\lambda^{(1)} + \mathcal{O}(N^{-2})\right),$$
$$Z_m = Z_m^{(0)}\left(1 + \frac{1}{N}\widehat{Z}_m^{(1)} + \mathcal{O}(N^{-2})\right), \quad (4.59)$$

and their first terms are equal:

$$Z_\lambda^{(0)} = Z_m^{(0)}. \quad (4.60)$$

A first consistency check of (4.58) is that it is finite, i.e. that the divergences in the diagrams of Fig. 6 cancel against the renormalization constants. To verify that, and to calculate the finite part, we will calculate the sum over $n$ appearing here by using the formalism explained in Appendix A.2. The structure function which calculates the sum

$$-\sum_{n\geq 1}\left(\mathrm{i}\mathcal{I}_1(n) + \frac{1}{2}(\mathrm{i}\pi\Pi)^n\right)\lambda_0^n \quad (4.61)$$

is given by

$$H(x,y) = -\frac{1}{2}\left(-\frac{p^2}{4\pi\nu^2}\right)^{-y/2}\left[\frac{\Gamma\left(1 + \frac{x}{2}\right)\Gamma^2\left(1 - \frac{x}{2}\right)}{\Gamma(1 - x)}\right]^{y/x-1}$$
$$\times\left[\frac{\Gamma\left(1 + \frac{y}{2}\right)\Gamma\left(1 - \frac{y}{2}\right)\Gamma\left(-\frac{x}{2}\right)}{\Gamma\left(\frac{y-x}{2}\right)\Gamma\left(1 - \frac{y+x}{2}\right)} - y\frac{\Gamma\left(1 + \frac{x}{2}\right)\Gamma\left(1 - \frac{x}{2}\right)\Gamma\left(-\frac{x}{2}\right)}{2\Gamma(1 - x)}\right]. \quad (4.62)$$

In particular, we have

$$H_0(\epsilon) = H(\epsilon, 0) = \frac{\chi(-\epsilon)}{\epsilon} - F_0(\epsilon). \quad (4.63)$$

In this expression, $F_0(\epsilon)$ is the function defined in (4.13) and appearing in the calculation of the perturbative self-energy, and $\chi(\lambda)$ is the function (2.22) in the mass anomalous dimension. Moreover, the $1/N$ expansion of the renormalization constants gives

$$\widehat{Z}_\lambda^{(1)} - \widehat{Z}_m^{(1)} = \int_0^\lambda \mathrm{d}u\frac{\chi(u)}{u(u + \epsilon)}. \quad (4.64)$$

Using (A.19) and (4.12), we obtain

$$\left[H_0(\epsilon)\log\left(1 + \frac{\lambda}{\epsilon}\right)\right]_{\mathrm{div}} = -\widehat{Z}_\lambda^{(1)} + \widehat{Z}_m^{(1)} - Z_\psi^{(1)}. \quad (4.65)$$

Now it is clear that all the divergent parts cancel in (4.58). In fact, one can use this calculation to determine the function $\chi(u)$, which essentially gives the mass anomalous dimension at NLO in the $1/N$ expansion.

Let us now evaluate the finite part of the sum. As shown in (A.23), it has two pieces. One of them can be read from the structure function evaluated at $x = 0$:

$$H(0, y) = \frac{y}{4}\left(2\gamma_E + \psi\left(1 - \frac{y}{2}\right) + \psi\left(\frac{y}{2}\right)\right). \tag{4.66}$$

After expanding the above expression in powers of $y$, we can extract the coefficients $H_m(0)$ and obtain

$$-\frac{1}{2} + \sum_{m \geq 1}(m - 1)! H_m(0)\lambda^m = -\frac{1}{2} - \sum_{k \geq 1}\frac{(2k)!}{2^{2k+1}}\zeta(2k + 1)\lambda^{2k+1}, \tag{4.67}$$

where we have included the term $-1/2$ originating from the diagram in Fig. 3b. The other finite piece combines with the non-trivial power series in $\lambda$ which appears in (4.30), when we express the two-quark condensate in terms of $\Lambda$, to produce

$$-\int^{\lambda}\frac{\chi(u)}{u^2}\mathrm{d}u - \int_0^{\lambda}\frac{H_0(-u) - H_{0,0}}{u}\mathrm{d}u = -\frac{1}{2}\log(\lambda) + \int_0^{\lambda}\frac{\gamma^{(1)}(u)}{\beta_\lambda^{(0)}(u)}\mathrm{d}u. \tag{4.68}$$

We finally obtain

$$\begin{aligned}\Sigma_{m,R}^{\mathrm{NP}} = c(N)\Lambda\bigg\{&1 - \frac{1}{N}\left(\frac{1}{2} + \frac{1}{2}\log(\lambda) + \sum_{k \geq 1}\frac{(2k)!}{2^{2k+1}}\zeta(2k + 1)\lambda^{2k+1}\right) \\ &+ \frac{1}{N}\int_0^{\lambda}\frac{\gamma^{(1)}(u)}{\beta_\lambda^{(0)}(\lambda)}\mathrm{d}u + \mathcal{O}(N^{-2})\bigg\}.\end{aligned} \tag{4.69}$$

The term in the second line is due to the change of scheme, since we have

$$m + \frac{m}{N}\Sigma_m^{\mathrm{SM}} = \Sigma_{m,R}\left(1 - \frac{1}{N}\int_0^{\lambda}\frac{\gamma^{(1)}(u)}{\beta_\lambda^{(0)}(u)}\mathrm{d}u + \mathcal{O}(N^{-2})\right). \tag{4.70}$$

Note that the correction involving the anomalous dimension of the field now comes with a minus sign, as compared to (4.21). We remove this scheme dependent term, write $\Lambda$ in terms of $m$ using the relation (3.15), and expand the factor $c(N)$ using (4.33). We finally obtain, from (4.69),

$$-\Sigma_m^{\mathrm{SM}} \sim 1 - \frac{\gamma_E}{2} + \log(2) - c_1 + \frac{1}{2}\log(\lambda) + \sum_{k \geq 1}\frac{(2k)!}{2^{2k+1}}\zeta(2k + 1)\lambda^{2k+1} + \mathcal{O}(N^{-1}). \tag{4.71}$$

Due to the second equation in (3.13), the series in the r.h.s. should be equal to the series $\Phi_1(\lambda)$ in (3.50). This is indeed the case for the $\lambda$-dependent terms. Equality of the constant term fixes the value of $c_1$:

$$c_1 = \frac{1}{2} + \frac{3}{2}\log(2). \tag{4.72}$$

The calculation above also fixes that $c(N) \approx 1$ at large $N$. We have already incorporated this information by using the large $N$ calculation of the condensate, but we could have obtained it from the comparison of our result (4.69) with the explicit result for the trans-series.

Let us note that our calculation of the two-quark condensate correction to the fermion self-energy is conceptually very similar to what has been done for QCD in [10, 41, 42]. One of the goals of such a calculation in QCD is to dynamically generate a mass for the quarks out of the condensate. We know from the large $N$ analysis of [16] that this is indeed the case in the Gross–Neveu model: the non-zero vev of $\sigma$ gives simultaneously a vev to the two-quark condensate and a mass to the fermions. However, these two quantities are conceptually different (e.g. the first one is not RG invariant, while the second one is), and this difference makes itself manifest at next-to-leading order in the $1/N$ expansion. Our analysis of the self-energy shows in detail how the contribution of the two-quark condensate to the OPE of the fermion propagator generates a mass pole at large $N$ and an additional power correction at order $1/N$, in the manner intended in QCD.

## 4.4 Trans-series for the four-quark condensate

We will now consider the contribution of the four-quark condensate to the trans-series. In the calculation of the two-point function there are two possible sources for such condensate. First, we can leave two pairs of fermions uncontracted when expanding the action. This leads to factors of $\langle V \rangle_c$. Second, we can consider diagrams with a single pair of uncontracted fermions, like the ones we studied in the previous section, but at different locations. When we expand them as in (4.52) we will get factors of $\langle K \rangle_c$. Although this operator is strictly speaking not a four-quark operator, its vev gives the same contribution, but with an opposite sign, as we saw in (4.27). We will then refer to these contributions as also due to a four-quark condensate, by a slight abuse of language.

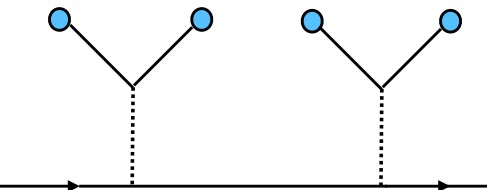

**Figure 7**: Diagram of order $g_0^2$ and order 1 in the $1/N$ expansion that contributes to the four-quark condensate correction to the two-point function.

In order to proceed, we have to understand how to extract the self-energy from the two-point function, as we did in the case of the two-quark condensate. To see how this goes, let us consider the possible diagrams with a four-quark condensate which contribute to the two-point function. It is easy to see that there is a diagram, and only one, which gives a contribution of order one in the $1/N$ expansion, and shown in Fig. 7 (the condensates come from quark pairs in the same vertices, so they give a factor of $N^2$, and the diagram goes like $g_0^2 N^2$). A precise evaluation of the diagram gives

$$\frac{g_0 \langle V \rangle_c}{p^2}. \tag{4.73}$$

Note that this does not factorize, so the diagram in Fig. 7 should not be regarded as a reducible diagram. The factorization only takes place at large $N$. In order to do the precise large $N$ counting, let us already renormalize the contribution of this diagram. For the composite operator $V$, we need the renormalization constant in (4.38):

$$Z_V = 1 - \frac{\beta_\lambda(\lambda)}{\lambda \epsilon} = Z_V^{(0)} \left( 1 + \frac{1}{N} \widehat{Z}_V^{(1)} + \mathcal{O}(N^{-2}) \right) \tag{4.74}$$

and we have

$$Z_V^{(0)} Z_\lambda^{(0)} = 1, \qquad \widehat{Z}_V^{(1)} = -\frac{\beta_\lambda^{(1)}(\lambda)}{\lambda(\lambda + \epsilon)}. \tag{4.75}$$

Using in addition the renormalization constant of the coupling and its $1/N$ expansion in (4.59), we find

$$Z_\lambda Z_V \frac{\pi\lambda}{N} \langle [V] \rangle_c = \frac{\pi\lambda}{N} \langle [V] \rangle_c \left( 1 + \frac{1}{N} \left( \widehat{Z}_\lambda^{(1)} + \widehat{Z}_V^{(1)} \right) + \cdots \right). \tag{4.76}$$

Note that we have the large $N$ scaling $\langle [V] \rangle_c \sim N$, so the front factor is of order 1. We will denote the renormalized two-quark and four-quark condensates by

$$\mathcal{T} = -\frac{\pi\lambda}{N} \langle [\overline{\psi}\psi] \rangle_c, \qquad \mathcal{F} = \frac{\pi\lambda}{N} \langle [V] \rangle_c. \tag{4.77}$$

By the large $N$ factorization of (4.26), one has

$$\mathcal{F} - \mathcal{T}^2 = \mathcal{O}(N^{-1}). \tag{4.78}$$

We also note that

$$\Sigma_{m,R}^{\mathrm{NP},0} = \mathcal{T}. \tag{4.79}$$

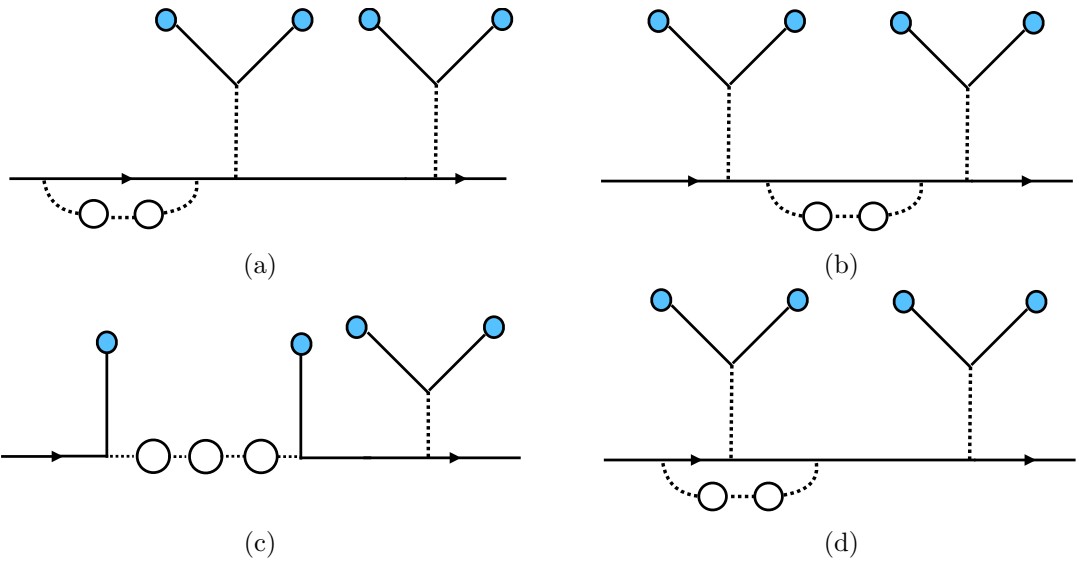

**Figure 8**: Diagrams with four-quark condensates that contribute to the two-point function but not to the self-energy, at order $1/N$. One should include as well the diagrams obtained by left-right reflection.

In addition to the diagram of Fig. 7, which is of order one at large $N$, we find many other diagrams of order $1/N$ in the calculation of the four-quark condensate corrections. Among these, there are the diagrams shown in Fig. 8 (together with their right-left reflections). These diagrams do not contribute to the self-energy, up to order $1/N$. To see this, we note that the term of order $\mathcal{C}^2$ in the expansion of the renormalized two-point function (4.50) is, up to order $1/N$,

$$\frac{1}{N} \Sigma_{p,R}^{\mathrm{NP},1} + \frac{1}{p^2} \left( \mathcal{T}^2 + \frac{2}{N} \mathcal{T} \Sigma_{m,R}^{\mathrm{NP},1} \right) + \frac{3}{p^2 N} \mathcal{T}^2 \Sigma_{p,R}^{\mathrm{P},1}. \tag{4.80}$$

To compare this expression with the diagrammtic computation, we still need to include the field renormalization constant and write renormalized quantities together with their original divergences. This yields

$$\frac{1}{N}\Sigma_{p,R}^{\mathrm{NP},1} + \frac{1}{p^2}\mathcal{T}^2 + \frac{2}{p^2 N}\mathcal{T}^2\left(\widehat{Z}_\lambda^{(1)} - \widehat{Z}_m^{(1)}\right)$$
$$+ \frac{3}{p^2 N}\mathcal{T}^2\left(\Sigma_{p,R}^{\mathrm{P},1} + Z_\psi^{(1)}\right) + \frac{2}{p^2 N}\mathcal{T}\left(\Sigma_{m,R}^{\mathrm{NP},1} + \mathcal{T}\left(-\widehat{Z}_\lambda^{(1)} + \widehat{Z}_m^{(1)} - Z_\psi^{(1)}\right)\right). \quad (4.81)$$

Due to the factorization (4.78), the first term in the second line corresponds to the diagrams in Fig. 8a–8b, while the second term in the second line corresponds to the diagrams in Fig. 8c–8d (in both cases, up to this order in $1/N$). Only the first line remains to be accounted for in (4.81), which has to be given by the diagram in Fig. 7 plus irreducible diagrams of order $1/N$, i.e. those that are not included in Fig. 8. Including the renormalization constants in (4.76), we obtain

$$\frac{1}{N}\Sigma_{p,R}^{\mathrm{NP}} = \frac{\mathcal{F} - \mathcal{T}^2}{p^2} + \frac{\mathcal{F}}{p^2 N}\left(\widehat{Z}_\lambda^{(1)} + \widehat{Z}_V^{(1)} - 2\left(\widehat{Z}_\lambda^{(1)} - \widehat{Z}_m^{(1)}\right)\right) - i\slashed{p} \times \text{irreducible} + \mathcal{O}\left(N^{-2}\right). \quad (4.82)$$

The first term in (4.82) can be written more explicitly by reexpressing the condensates in terms of $\Lambda$, through (4.30) and (4.34). We find

$$\mathcal{F} - \mathcal{T}^2 = \frac{\Lambda^2}{N}\left(d_1 - 2c_1 + \log(\lambda) - 1 + \frac{\beta_\lambda^{(1)}(\lambda)}{\lambda^2} + 2\int_0^\lambda \frac{\chi(u) - \chi_1 u}{u^2}\mathrm{d}u\right) + \mathcal{O}\left(N^{-2}\right), \quad (4.83)$$

where the coefficients $d_1$, $c_1$ were introduced in (4.33), (4.35). Based on the general arguments in [2, 3], we expect $d_1$, the $1/N$ correction to the four-quark condensate, to be ambiguous. We will see that this is indeed the case.

Let us note that the second term in the r.h.s. of (4.82) should cancel the divergences obtained from these diagrams. Let us find a more convenient form for this combination of renormalization constants. We first derive from (2.28) the integral form

$$\widehat{Z}_\lambda^{(1)} = \int_0^\lambda \frac{\epsilon\beta_\lambda^{(1)}(u)}{u^2(u+\epsilon)^2}\mathrm{d}u. \quad (4.84)$$

After integration by parts we obtain the convenient expression

$$\widehat{Z}_\lambda^{(1)} + \widehat{Z}_V^{(1)} = \int_0^\lambda \frac{\rho(u)}{u(u+\epsilon)}\mathrm{d}u, \quad (4.85)$$

where

$$\rho(\lambda) = -\lambda^2\frac{\mathrm{d}}{\mathrm{d}\lambda}\left(\frac{\beta_\lambda^{(1)}(\lambda)}{\lambda^2}\right). \quad (4.86)$$

The quantity $\widehat{Z}_\lambda^{(1)} - \widehat{Z}_m^{(1)}$ is known from the calculation of the two-quark condensate (4.64). Therefore, the singular part in the irreducible diagrams that contribute to the four-quark condensate determines the beta function at NLO in $1/N$. As we will see, this calculation is simpler than the one usually adopted for the calculation of the beta function at this order in [18, 19]. By using the known value (2.18), we have

$$\rho(\lambda) = -\frac{\lambda^2\Gamma(2+\lambda)}{(2+\lambda)\Gamma\left(1-\frac{\lambda}{2}\right)\Gamma^3\left(1+\frac{\lambda}{2}\right)} = -\lambda\chi(\lambda). \quad (4.87)$$

We can now write

$$\widehat{Z}_\lambda^{(1)} + \widehat{Z}_V^{(1)} - 2\left(\widehat{Z}_\lambda^{(1)} - \widehat{Z}_m^{(1)}\right) = \int_0^\lambda \frac{v(u)}{u(u+\epsilon)}\mathrm{d}u, \tag{4.88}$$

where

$$v(u) = -\frac{u\Gamma(2+u)}{\Gamma^3\left(1+\frac{u}{2}\right)\Gamma\left(1-\frac{u}{2}\right)}. \tag{4.89}$$

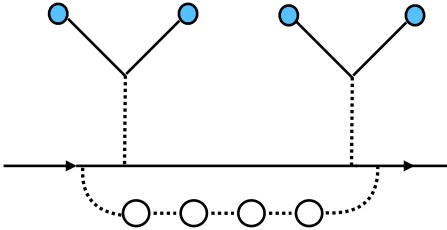

**Figure 9**: Irreducible diagram that contributes to the four-quark condensate.

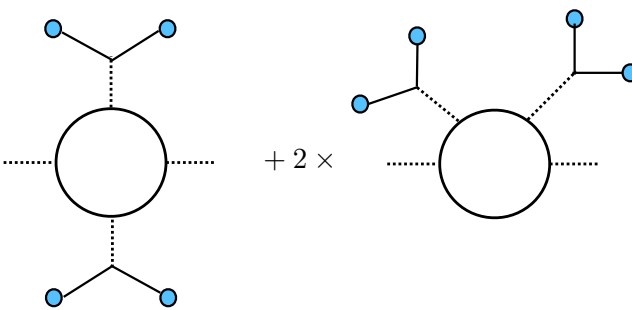

**Figure 10**: Two possible ways of inserting a four-quark condensate in a fermion polarization loop. The second diagram needs a symmetry factor 2.

We have then to find the irreducible diagrams, to order $1/N$ and all loops. There are four types of diagrams that contribute:

1. The first type of diagram is obtained by putting a four-quark condensate in the middle of the internal propagator line of the bubble chain in Fig. 2. This gives the diagram shown in Fig. 9.

2. The second type of diagram is obtained by noticing that one can insert a four-quark condensate in a fermion polarization loop, in two different ways, to obtain a "decorated" polarization loop, as shown in Fig. 10. This "decorated" loop can then be inserted at any point inside a bubble chain, and leads to diagrams like the one in Fig. 11.

3. The third type of diagram is obtained by noticing that one can insert a four-quark condensate in the sigma propagator, which can then be inserted in a bubble chain, as shown in Fig. 12.

4. Finally, there are contributions coming from diagrams involving a two-quark condensate, in which the quark and antiquark are at different points, and one has to expand. The diagrams that contribute to this are the diagram in Fig. 6a, and the diagram in Fig. 13.

Note that the latter gives a vanishing contribution to the two-quark condensate, but not to the four-quark condensate.

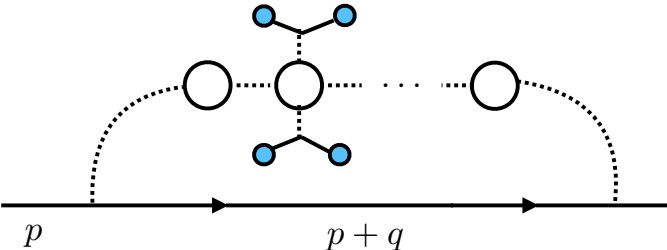

**Figure 11**: A chain of bubbles with an insertion of one of the "decorated" polarization loops.

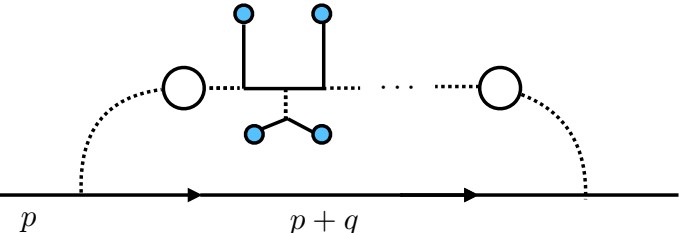

**Figure 12**: A chain of bubbles with an insertion of a four-quark condensate in the propagator of the sigma field.

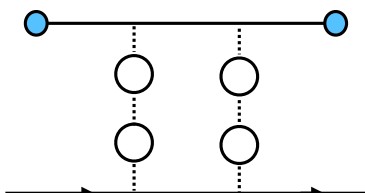

**Figure 13**: When we expand the two-quark condensate around the same point, this diagram gives a contribution to the four-quark condensate.

Let us now compute the contribution of the irreducible diagrams. It will be convenient to group them in appropriate ways.

We first consider the irreducible diagram in Fig. 9, which we will combine with the diagram in Fig. 6a. The contribution of Fig. 9 to $\Sigma_p^{\mathrm{NP}}$ is of the form

$$-\frac{\pi\lambda_0}{N^2}\frac{\langle V\rangle_c}{p^2}\sum_{n\geq 1}\mathrm{i}\mathcal{I}_2(n)\lambda_0^n,\tag{4.90}$$

where

$$\mathcal{I}_2(n)=\frac{p^2}{\not{p}}\pi^n\int\frac{\mathrm{d}^dq}{(2\pi)^d}\frac{\not{p}-\not{q}}{(p-q)^4}\left(\mathrm{i}\Pi(q^2)\right)^{n-1}.\tag{4.91}$$

Let us now consider the diagram in Fig. 6a. In the previous section we calculated the first term in the expansion in the first line of (4.52), and we have to consider now the second term,

which will produce a factor of $\langle K \rangle_c$. Its contribution to the diagram in position space involves an integral of the form

$$\int d^d y_1 d^d y_2 (y_1 - y_2)^\rho e^{i(p-q)y_1 + i(q-k)y_2} \left( \Pi(q^2) \right)^n \tag{4.92}$$

where $p$ is the external momentum, $k$ and $q$ are internal momenta to be integrated over, and $n$ is the number of inserted bubbles. To calculate this integral, we write

$$(y_1 - y_2)^\rho e^{iq(y_2 - y_1)} = -\frac{1}{i} \frac{\partial}{\partial q_\rho} e^{iq(y_2 - y_1)} \tag{4.93}$$

and we integrate by parts (see e.g. [40, 42] for similar calculations). After taking everything into account, we find that this diagram gives a contribution to $\Sigma_p^{\mathrm{NP}}$ of the form

$$-\frac{\pi \lambda_0}{dN^2} \langle K \rangle_c \sum_{n \geq 1} \frac{\partial (i\pi \Pi(p^2))^n}{\partial p^2} \lambda_0^n, \tag{4.94}$$

which will combine with (4.90) into

$$\frac{\pi \lambda_0}{N^2} \frac{\langle V \rangle_c}{p^2} \sum_{n \geq 1} \left( \frac{p^2}{d} \frac{\partial (i\pi \Pi(p^2))^n}{\partial p^2} - i\mathcal{I}_2(n) \right) \lambda_0^n, \tag{4.95}$$

after using (4.27). The general formula (A.3) gives

$$\mathcal{I}_2(n) = i \left( -\frac{p^2}{4\pi} \right)^{-n\epsilon/2} \frac{\Gamma\left(-\frac{\epsilon}{2}\right)\Gamma\left(1 + \frac{n\epsilon}{2}\right)\Gamma\left(1 - \frac{n\epsilon}{2}\right)}{\Gamma\left(\frac{(n-1)\epsilon}{2}\right)\Gamma\left(1 - \frac{(n+1)\epsilon}{2}\right)} \left[ -\frac{1}{2} \frac{\Gamma\left(1 + \frac{\epsilon}{2}\right)\Gamma\left(1 - \frac{\epsilon}{2}\right)\Gamma\left(-\frac{\epsilon}{2}\right)}{\Gamma(1 - \epsilon)} \right]^{n-1}. \tag{4.96}$$

On the other hand,

$$\frac{p^2}{d} \frac{\partial (i\pi \Pi(p^2))^n}{\partial p^2} = -\frac{1}{4} \left( -\frac{p^2}{4\pi} \right)^{-n\epsilon/2} \left( -\frac{n\epsilon}{2} \right) \frac{\Gamma\left(1 + \frac{\epsilon}{2}\right)\Gamma\left(1 - \frac{\epsilon}{2}\right)\Gamma\left(-\frac{\epsilon}{2}\right)}{\left(1 - \frac{\epsilon}{2}\right)\Gamma(1 - \epsilon)}$$
$$\times \left[ -\frac{1}{2} \frac{\Gamma\left(1 + \frac{\epsilon}{2}\right)\Gamma\left(1 - \frac{\epsilon}{2}\right)\Gamma\left(-\frac{\epsilon}{2}\right)}{\Gamma(1 - \epsilon)} \right]^{n-1}. \tag{4.97}$$

Combining both, and using the technique of Appendix A.2, we find that the sum in (4.95) is governed by the structure function

$$M(x, y) = -\frac{1}{4} \left( -\frac{p^2}{4\pi\nu^2} \right)^{-y/2} \left[ \frac{\Gamma\left(1 + \frac{x}{2}\right)\Gamma^2\left(1 - \frac{x}{2}\right)}{\Gamma(1 - x)} \right]^{y/x - 1}$$
$$\times y \left[ \frac{\Gamma\left(-\frac{x}{2}\right)\Gamma\left(1 + \frac{y}{2}\right)\Gamma\left(1 - \frac{y}{2}\right)}{\Gamma\left(\frac{y-x}{2}\right)\Gamma\left(1 - \frac{y+x}{2}\right)} - \frac{y}{2} \frac{\Gamma\left(1 + \frac{x}{2}\right)\Gamma\left(1 - \frac{x}{2}\right)\Gamma\left(-\frac{x}{2}\right)}{\left(1 - \frac{x}{2}\right)\Gamma(1 - x)} \right]. \tag{4.98}$$

Note also that $M_0(x) = 0$. This means that the sum (4.95) can be made finite simply by renormalizing the coupling constant. We also find

$$M(0, y) = \frac{y^2}{8} \left( -1 + 2\gamma_E + \psi\left(1 - \frac{y}{2}\right) + \psi\left(\frac{y}{2}\right) \right). \tag{4.99}$$

We conclude that the contribution of these two classes of diagrams to $\Sigma_p^{\mathrm{NP}}$ is

$$-\frac{m^2}{p^2}\left\{\frac{\lambda}{4}+\frac{\lambda^2}{8}+\sum_{k\geq 1}\frac{(2k+1)!}{2^{2k+2}}\zeta(2k+1)\lambda^{2k+2}\right\}.\tag{4.100}$$

We will now consider the combination of the diagrams in Fig. 11 and Fig. 12. First of all, we calculate the amplitude associated to the "decorated" loops in Fig. 10. We find

$$\Pi_{(\bar{\psi}\psi)^2}(p^2)=2\int\frac{\mathrm{d}^d q}{(2\pi)^d}\left[\frac{3}{q^2(q-p)^2}-\frac{2q\cdot p}{q^4(q-p)^2}\right]=-\frac{2}{p^2}(1-\epsilon)\Pi(p^2).\tag{4.101}$$

Here we have included both the sign $-1$ due to the fermionic loop and the $-1=(-\mathrm{i})^2$ coming from the two $\sigma$ propagators. The symmetry factor of the diagram in Fig. 11, in which there are $n-2$ conventional polarization loops and one decorated loop, is $n-1$. The diagram in Fig. 12, in which there are $n-1$ polarization bubbles, has a symmetry factor $2n$. The contribution of these two classes of diagrams to the self-energy is

$$\frac{\pi\lambda_0}{N^2}\frac{\langle V\rangle_c}{p^2}\sum_{n\geq 1}(-2n\mathrm{i}\mathcal{I}_3(n)-(n-1)\mathrm{i}\mathcal{I}_4(n))\lambda_0^n,\tag{4.102}$$

where

$$\mathcal{I}_3(n)=\frac{p^2}{\not p}\pi^n\int\frac{\mathrm{d}^d q}{(2\pi)^d}\frac{\not p-\not q}{q^2(p-q)^2}\left(\mathrm{i}\Pi(q^2)\right)^{n-1}\tag{4.103}$$

is the integral associated to the diagram in Fig. 12, while

$$\mathcal{I}_4(n)=\frac{p^2}{\not p}\pi^n\int\frac{\mathrm{d}^d q}{(2\pi)^d}\frac{\not p-\not q}{(p-q)^2}\left(\mathrm{i}\Pi(q^2)\right)^{n-2}\left(\mathrm{i}\Pi_{(\bar{\psi}\psi)^2}(q^2)\right)\tag{4.104}$$

is the integral associated to the diagram in Fig. 11. In view of (4.101), the two integrals are related as

$$\mathcal{I}_4(n)=-2(1-\epsilon)\mathcal{I}_3(n).\tag{4.105}$$

The integral $\mathcal{I}_3(n)$ can be computed with the expression (A.3), and we find that the sum in (4.102) is governed by the structure function

$$R(x,y)=\left(-\frac{p^2}{4\pi\nu^2}\right)^{-y/2}(1+y-x)\frac{\Gamma\left(1-\frac{x}{2}\right)\Gamma\left(1-\frac{y}{2}\right)\Gamma\left(1+\frac{y}{2}\right)}{\Gamma\left(1+\frac{y-x}{2}\right)\Gamma\left(1-\frac{y+x}{2}\right)}\left[\frac{\Gamma\left(1+\frac{x}{2}\right)\Gamma^2\left(1-\frac{x}{2}\right)}{\Gamma(1-x)}\right]^{y/x-1}.\tag{4.106}$$

We note that

$$R_0(x)=\frac{\Gamma(2-x)}{\Gamma^3\left(1-\frac{x}{2}\right)\Gamma\left(1+\frac{x}{2}\right)}=\frac{\upsilon(-x)}{x},\tag{4.107}$$

where $\upsilon(x)$ was introduced in (4.89). We also have

$$R(0,y)=1+y\tag{4.108}$$

for $\mu^2=-p^2$. Let us calculate the finite and divergent parts due to (4.107). We have

$$\left[R_0(\epsilon)\log\left(1+\frac{\lambda}{\epsilon}\right)\right]_{\mathrm{div}}=\int_0^\lambda\frac{R_0(-u)}{u+\epsilon}\mathrm{d}u=-\int_0^\lambda\frac{\upsilon(u)}{u(u+\epsilon)}\mathrm{d}u.\tag{4.109}$$

This will cancel precisely the divergent part in (4.88). Therefore, the diagrams of Fig. 11 and Fig. 12 are the relevant ones to compute the anomalous dimension of the operator $(\overline{\psi}\psi)^2$ and, therefore, of the beta function at NLO. A related calculation of this anomalous dimension in [44] uses these diagrams in disguise.

The finite part due to $R_0(x)$ is given by

$$-\int_0^\lambda \frac{R_0(-u)-1}{u}\mathrm{d}u = 1 - \frac{\beta_\lambda^{(1)}(\lambda)}{\lambda^2} - 2\int_0^\lambda \frac{\chi(u)-\chi_1 u}{u^2}\mathrm{d}u, \tag{4.110}$$

and it cancels with part of the expression in (4.83), leaving us with the following contribution to $\Sigma_{p,R}^{\mathrm{NP}}$:

$$\frac{m^2}{p^2}(d_1 - 2c_1 + \log(\lambda)). \tag{4.111}$$

This result combines with the finite part coming from (4.108), resulting in the contribution

$$\frac{m^2}{p^2}(d_1 - 2c_1 + \log(\lambda) + \lambda). \tag{4.112}$$

Let us finally consider the contribution of the family of diagrams in Fig. 13. When the two quarks are at the same point, this diagram is proportional to $\mathrm{Tr}(\gamma^\mu)$ and it vanishes. When we expand the two quark fields around the same point, we find a contribution to the four-quark condensate which involves an integral of the form

$$\int \mathrm{d}^d y_1 \mathrm{d}^d y_2\, \mathrm{e}^{\mathrm{i}(p-k-q)y_1 - \mathrm{i}(r-k-q)y_2}(y_1-y_2)^\rho \frac{\mathrm{i}}{\not{k}}\frac{q_\rho}{q^2}, \tag{4.113}$$

where $p$ is the external momentum and $k, q, r$ are internal momenta to be integrated over. In the integration by parts one obtains

$$\frac{\partial}{\partial q_\rho}\left(\frac{q_\rho}{q^2}\right) = \frac{d-2}{q^2}. \tag{4.114}$$

The contribution of the sum over all bubbles is

$$-\frac{\pi\lambda_0}{N^2}\frac{\langle V\rangle_c}{p^2}\sum_{n\geq 1}\frac{d-2}{d}2n\mathrm{i}\mathcal{I}_3(n)\lambda_0^n, \tag{4.115}$$

which can be calculated in terms of the structure function

$$S(x,y) = -\left(-\frac{p^2}{4\pi\nu^2}\right)^{-y/2}\frac{y}{2-x}\frac{\Gamma\left(1-\frac{x}{2}\right)\Gamma\left(1-\frac{y}{2}\right)\Gamma\left(1+\frac{y}{2}\right)}{\Gamma\left(1-\frac{y+x}{2}\right)\Gamma\left(1+\frac{y-x}{2}\right)}\left[\frac{\Gamma\left(1+\frac{x}{2}\right)\Gamma^2\left(1-\frac{x}{2}\right)}{\Gamma(1-x)}\right]^{y/x-1}. \tag{4.116}$$

We find $S_0(x) = 0$, and

$$S(0,y) = -\frac{y}{2}. \tag{4.117}$$

Including the factors in front of the sum in (4.115), we obtain a contribution to $\Sigma_{p,R}^{\mathrm{NP}}$ of the form

$$-\frac{m^2}{p^2}\frac{\lambda}{2}. \tag{4.118}$$

The results in (4.100), (4.112) and (4.118) combine into

$$\Sigma_{p,R}^{\text{NP}} = -\frac{m^2}{p^2}\left\{2c_1 - d_1 - \log(\lambda) - \frac{\lambda}{4} + \frac{\lambda^2}{8} + \sum_{k\geq 1}\frac{(2k+1)!}{2^{2k+2}}\zeta(2k+1)\lambda^{2k+2}\right\} + \mathcal{O}(N^{-1}). \quad (4.119)$$

This agrees precisely with the result in (3.51), involving the series $\Phi_2(\lambda)$ of (3.47), except for the constant terms, which depend on the values of the condensates. Moreover, by comparing the two results we can read off the value of $d_1$, giving the $1/N$ correction to the four-quark condensate

$$d_1 = 1 - \gamma_E + 2\log(2) \mp \frac{\mathrm{i}\pi}{2}. \quad (4.120)$$

As advertised, this is ambiguous due to the renormalon in the perturbative series, and the choice of sign in (4.120) should be correlated with a choice of resummation prescription for the series $\Phi_0(\lambda)$.

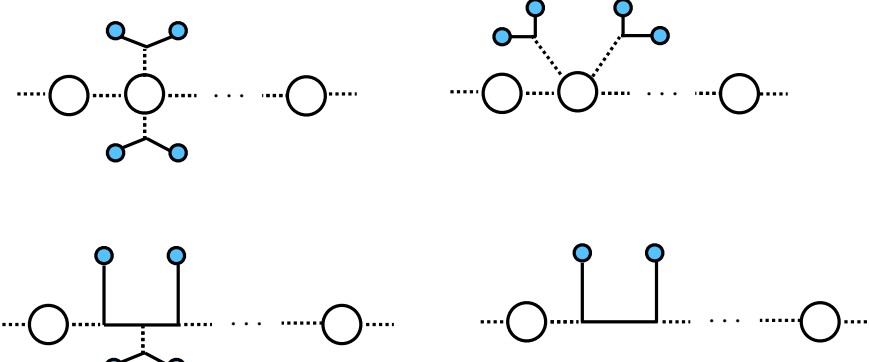

**Figure 14**: The diagrams contributing to the four-quark condensate correction to the sigma propagator, at order $1/N$.

A simple corollary of this calculation is a determination of the four-quark condensate contribution to the propagator of the sigma particle, at order $1/N$. Diagrammatically it is given by the sum of bubbles with condensate insertions shown in Fig. 14. By using the results (4.102) and (4.115), one finds

$$-\frac{2\pi\mathrm{i}}{N}\frac{2m_0^2}{p^2}\left(\frac{1}{\log(-p^2/m_0^2)} + \frac{1}{\log^2(-p^2/m_0^2)}\right). \quad (4.121)$$

It is easy to check that this agrees with the result of expanding (3.5) at large $p^2 \gg m_0^2$.

## 5 Conclusions and open problems

The combination of the OPE with vacuum condensates, as developed in the SVZ sum rules, produces formal trans-series for QFT observables, and it provides a method to systematically calculate exponentially small corrections to perturbative quantities. In this paper we have compared the result of an OPE calculation to an exact trans-series obtained in the $1/N$ expansion, and we have found complete agreement between the two, up to the unknown values of the condensates.

In our calculation we have used the "practical" version of the OPE. As we mentioned in the Introduction, it has been suggested that one should use a more complicated version, based on the introduction of an additional momentum scale. In this "Wilsonian" version, IR renormalons are absent, since the additional scale provides an explicit IR cutoff for the Feynman integrals, and the condensates are defined unambiguously. The prize to pay is that each series in the trans-series depends on this scale, and the dependence only drops out in the total result. Our study of the Gross–Neveu model seems to confirm that there is nothing wrong with the "practical" version of the OPE, in agreement with the discussions in [3, 45]. The exact results for the two-point function can be decoded in terms of a trans-series in which condensates are ambiguous, but this ambiguity is due to the well-known Stokes phenomenon and does not lead to any inconsistency. In addition, the series in the 't Hooft parameter appearing in this trans-series can be reproduced exactly, and rather non-trivially, with the "practical" OPE.

A nice outcome of our calculation is the following. In the exact large $N$ answer, the trans-series emerges as a formal, algebraic object. The OPE calculation gives a concrete picture of this trans-series in terms of perturbation theory with condensates. In particular, the facto-rial divergence of the perturbative series appearing in the trans-series is the manifestation of a renormalon-like phenomenon, due to bubble chains attached to the condensates, as illustrated in e.g. Fig. 6 or Fig. 9.

Although this is probably well-known to many practitioners, it is worth noting that the OPE calculation reconstructs the trans-series more efficiently than a resurgent/renormalon analysis of the perturbative series. In particular, the two-quark condensate leads to a non-ambiguous power correction which is completely invisible in a resurgent analysis. The Borel singularity of the perturbative series detects the presence of a power correction in $m^2/p^2$ due to the four-quark condensate, but since this singularity is essentially a simple pole it misses the full series $\Phi_2(\lambda)$. The OPE calculation is in contrast able to reproduce the series $\Phi_{1,2}(\lambda)$ at all loops. It was suggested in [46] that part of the "blindness" of resurgent analysis in this situation might be due to the restriction to a given order in the $1/N$ expansion. It might happen that at finite $N$ one can have a better access to the four-quark condensate through the resurgent structure of the perturbative series, but the two-quark condensate series will remain undetected, since it transforms non-trivially under the $\mathbb{Z}_2$ chiral symmetry and does not mix with the identity operator [2]. In the language of [46], the trans-series for the self-energy at order $1/N$ does not satisfy the strong version of the resurgence program, since the resurgent analysis of the perturbative series does not make it possible to reconstruct the full trans-series. It does satisfy however the weak version of the program, since the exact result can be obtained by the (lateral) Borel resummation of the trans-series.

Our calculation can be extended in many ways. As we mentioned in the Introduction, the Gross–Neveu model turns out to be a simpler example to study than bosonic sigma models in two dimensions, since in the latter the fields are constrained and in addition there is an infinite number of operators with a fixed scaling dimension. It would be very interesting however to reproduce the large $N$ trans-series obtained in [1] (or the supersymmetric version studied more recently in [15]) by an OPE calculation with condensates similar to the one done here. The results of [14, 15] on non-linear sigma models might be a good starting point for this calculation.

Although going beyond the $1/N$ expansion is analytically difficult, we note that two-point functions can be computed non-perturbatively in integrable models by using form factors. In the case of the non-linear sigma model, it has been checked numerically that the form factor calculation reproduces asymptotically the perturbative series [47, 48]. It would be very interesting to see whether it is possible to detect as well condensate corrections to the perturbative result

through form factors.

Finally, although the results of this paper vindicate the idea that the OPE with vacuum condensates leads to the correct exponentially small corrections to the perturbative series, it is still not clear how the fractional power corrections found in [4] can be reproduced by a first principles calculation. This remains in our view a sharp open problem for our understanding of non-perturbative effects in QFT.

## Acknowledgements

We would like to thank Martin Beneke, Tomás Reis and Marco Serone for very useful discussions and a careful reading of the draft. The work of M.M. has been supported in part by the ERC-SyG project "Recursive and Exact New Quantum Theory" (ReNewQuantum), which received funding from the European Research Council (ERC) under the European Union's Horizon 2020 research and innovation program, grant agreement No. 810573. The work of R.M. is supported by the NKFIH K134946 Grant.

## A    Bubbles

### A.1    The fermion polarization loop

The building block of a bubble chain is the fermion polarization loop, given by the following diagram:

$$\qquad\qquad\qquad\qquad\qquad\qquad\qquad\qquad\qquad\qquad\qquad\qquad\text{(A.1)}$$

We define the corresponding amplitude, with massless fermions, as

$$\Pi(p^2) = \int \frac{\mathrm{d}^d q}{(2\pi)^d} \mathrm{Tr}\left[\frac{1}{\slashed{q}(\slashed{p}+\slashed{q})}\right] = 2\int \frac{\mathrm{d}^d q}{(2\pi)^d}\frac{q\cdot p}{q^2(p+q)^2}, \tag{A.2}$$

where we used dimensional regularization to drop a scale-less integral. This and other integrals appearing in this paper can be computed with the master formula

$$\int \frac{\mathrm{d}^d q}{(2\pi)^d} \frac{q^{(\mu_1\mu_2\dots\mu_n)}}{(q^2)^r[(p-q)^2]^s} = -\frac{\mathrm{i}}{4\pi}\left(-\frac{p^2}{4\pi}\right)^{-\epsilon/2}\frac{1}{(p^2)^{r+s-1}}p^{(\mu_1\mu_2\dots\mu_n)}$$
$$\times \frac{\Gamma(1+n-r-\epsilon/2)\Gamma(1-s-\epsilon/2)\Gamma(r+s-1+\epsilon/2)}{\Gamma(r)\Gamma(s)\Gamma(2-r-s+n-\epsilon)}, \tag{A.3}$$

where $q^{(\mu_1\mu_2\dots\mu_n)}$ is the traceless symmetric tensor constructed from $q^{\mu_1}q^{\mu_2}\dots q^{\mu_n}$, see Appendix C in [40] for more details. For example,

$$q^{(\mu)} = q^\mu, \qquad q^{(\mu_1\mu_2)} = q^{\mu_1}q^{\mu_2} - \frac{1}{d}g^{\mu_1\mu_2}q^2, \tag{A.4}$$

but we will only need to compute integrals with one index at most. Using (A.3), we find

$$\Pi(q^2) = \frac{\mathrm{i}}{2\pi}\left(-\frac{q^2}{4\pi}\right)^{-\epsilon/2}\frac{\Gamma\left(1+\frac{\epsilon}{2}\right)\Gamma\left(1-\frac{\epsilon}{2}\right)\Gamma\left(-\frac{\epsilon}{2}\right)}{\Gamma(1-\epsilon)}. \tag{A.5}$$

Let us now briefly consider the massive case. The massive polarization loop is defined as

$$\Pi_{m_0}(p^2) = \int \frac{\mathrm{d}^d q}{(2\pi)^d} \mathrm{Tr}\left[\frac{1}{(\slashed{q} - m_0)(\slashed{p} + \slashed{q} - m_0)}\right]. \tag{A.6}$$

Using standard Feynman integral techniques, one finds in $d = 2 - \epsilon$ dimensions the $\epsilon$ expansion (see e.g. [49] for additional details)

$$\Pi_{m_0}(p^2) = -\frac{\mathrm{i}}{\pi\epsilon} + \frac{\mathrm{i}}{2\pi}\log\left(\frac{m_0^2}{4\pi\mathrm{e}^{-\gamma_E}}\right) + \frac{\mathrm{i}}{2\pi}\xi\log\left[\frac{\xi+1}{\xi-1}\right] + \mathcal{O}(\epsilon), \tag{A.7}$$

where $\xi$ was introduced in (3.6). The $\sigma$ propagator in momentum space is closely related to the massive polarization loop:

$$\Delta^{-1}(p; m_0) = \Pi_{m_0}(p^2) - \frac{1}{m_0}\int \frac{\mathrm{d}^d q}{(2\pi)^d}\mathrm{Tr}\left[\frac{1}{\slashed{q} - m_0}\right], \tag{A.8}$$

and by using (A.7) one finds the expression (3.5).

## A.2 Summing over bubbles

In [18, 21], a powerful and simple technique was introduced to obtain the renormalized perturbative series associated to a chain of bubbles. We now summarize its main ingredients.

Let us consider a generic sum of corrections in the bare 't Hooft coupling $\lambda_0$, of order $1/N$:

$$A = \sum_{n \geq n_0} a_n(p^2)\lambda_0^n, \tag{A.9}$$

where $n_0 \geq 1$. The choice of $\lambda_0$ is such that, at leading order in the $1/N$ expansion, the renormalization function is given by (4.4). Since the quantity we are considering is already of order $1/N$, we only have to use the leading term $Z_\lambda^{(0)}$. Then the renormalized sum is of the form

$$A = \sum_{n \geq n_0} (\nu^2)^{n\epsilon/2}a_n(p^2)\left(Z_\lambda^{(0)}\right)^n\lambda^n. \tag{A.10}$$

Let us assume that one can find a "structure function" $F(x, y)$, which is analytic in both arguments at $x = 0$, $y = 0$, and satisfying

$$(\nu^2)^{n\epsilon/2}a_n(p^2) = \frac{F(\epsilon, n\epsilon)}{n\epsilon^n}. \tag{A.11}$$

Then the renormalized sum is of the form

$$A = \sum_{n \geq n_0} f_n\lambda^n\left(1 + \frac{\lambda}{\epsilon}\right)^{-n}, \tag{A.12}$$

where we have abbreviated

$$f_n = \frac{F(\epsilon, n\epsilon)}{n\epsilon^n}. \tag{A.13}$$

We now expand the factor $(1 + \lambda/\epsilon)^{-n}$ in powers of $\lambda$, using the binomial theorem, and we get

$$
A = \sum_{n \geq n_0} f_n \lambda^n \sum_{s \geq 0} \binom{n+s-1}{s}(-1)^s \left(\frac{\lambda}{\epsilon}\right)^s = \sum_{m \geq 1} \lambda^m \sum_{s=0}^{m-1} \binom{m-1}{s} \frac{(-1)^s}{\epsilon^s} f_{m-s}
$$

$$
= \sum_{m \geq n_0} \left(\frac{\lambda}{\epsilon}\right)^m \sum_{s=0}^{m-1} \binom{m-1}{s}(-1)^s \sum_{j \geq 0} F_j(\epsilon)(m-s)^{j-1} \epsilon^j \qquad \text{(A.14)}
$$

$$
= \sum_{m \geq n_0} \lambda^m \sum_{j \geq 0} \frac{F_j(\epsilon)}{\epsilon^{m-j}} \sum_{s=0}^{m-1} \binom{m-1}{s}(-1)^s (m-s)^{j-1},
$$

where we set $m = n + s$ and we performed the following expansion of the structure function:

$$
F(x,y) = \sum_{j \geq 0} F_j(x) y^j. \qquad \text{(A.15)}
$$

We now use that

$$
A = \sum_{s=0}^{m-1} \binom{m-1}{s}(-1)^s (m-s)^{j-1} = \begin{cases} \frac{(-1)^{m-1}}{m}, & \text{if } j = 0, \\ 0, & \text{if } 1 \leq j \leq m-1, \\ (m-1)!, & \text{if } j = m, \end{cases} \qquad \text{(A.16)}
$$

to write the renormalized sum as

$$
A = F_0(\epsilon) \sum_{m \geq n_0} \frac{(-1)^{m-1}}{m} \left(\frac{\lambda}{\epsilon}\right)^m + \sum_{m \geq n_0} (m-1)! F_m(\epsilon) \lambda^m + \mathcal{O}(\epsilon). \qquad \text{(A.17)}
$$

The first series can also be expressed as

$$
F_0(\epsilon) \sum_{m \geq n_0} \frac{(-1)^{m-1}}{m} \left(\frac{\lambda}{\epsilon}\right)^m = F_0(\epsilon) \log\left(1 + \frac{\lambda}{\epsilon}\right) - F_0(\epsilon) \sum_{m=1}^{n_0-1} \frac{(-1)^{m-1}}{m} \left(\frac{\lambda}{\epsilon}\right)^m. \qquad \text{(A.18)}
$$

We now consider the behavior as $\epsilon \to 0$. There is a divergent part encoded in (A.18). We note the useful formula

$$
\left[ F_0(\epsilon) \log\left(1 + \frac{\lambda}{\epsilon}\right) \right]_{\text{div}} = \int_0^\lambda \frac{F_0(-u)}{u + \epsilon} \mathrm{d}u, \qquad \text{(A.19)}
$$

and we can write the total divergent part in (A.18) as

$$
[A]_{\text{div}} = \int_0^\lambda \frac{F_0(-u)}{u + \epsilon} \mathrm{d}u - \sum_{m=1}^{n_0-1} \frac{(-1)^{m-1}}{m} \lambda^m \sum_{k=0}^{m-1} F_{0,k} \epsilon^{k-m}, \qquad \text{(A.20)}
$$

where the coefficients $F_{0,k}$ are defined by

$$
F_0(\epsilon) = \sum_{k \geq 0} F_{0,k} \epsilon^k. \qquad \text{(A.21)}
$$

The finite part of (A.17) arises from the second sum plus the terms of order $\epsilon^0$ in the first sum:

$$[A]_{\text{finite}} = \sum_{m \geq n_0} \frac{(-1)^{m-1}}{m} F_{0,m} \lambda^m + \sum_{m \geq n_0} (m-1)! F_m(\epsilon) \lambda^m. \tag{A.22}$$

It will be convenient to sum the first series into an integral, yielding

$$[A]_{\text{finite}} = -\int_0^\lambda \frac{F_0(-u) - F_{0,0}}{u} \mathrm{d}u - \sum_{m=1}^{n_0-1} \frac{(-1)^{m-1}}{m} F_{0,m} \lambda^m + \sum_{m \geq n_0} (m-1)! F_m(0) \lambda^m. \tag{A.23}$$

The last sum in the above expression has the form of an inverse Borel transform.

As we have seen, the structure function (A.11) is the relevant object in diagrammatic computations, as it contains all the necessary information to extract both the divergent part (A.20) and the finite part (A.23).

Another useful result for the calculation of renormalization functions is the following. Let $f(\epsilon)$ be analytic at $\epsilon = 0$. Then [17]

$$(\lambda + \epsilon) \frac{\partial}{\partial \lambda} \left[ \log\left(1 + \frac{\lambda}{\epsilon}\right) f(\epsilon) \right]_{\text{div}} = f(-\lambda). \tag{A.24}$$

This follows from a direct calculation:

$$
\begin{aligned}
(\lambda + \epsilon) \frac{\partial}{\partial \lambda} \left[ \log\left(1 + \frac{\lambda}{\epsilon}\right) f(\epsilon) \right]_{\text{div}} &= \sum_{m \geq 1} (-1)^{m-1} (\lambda^m + \epsilon \lambda^{m-1}) \sum_{k=0}^{m-1} f_k \epsilon^{k-m} \\
&= \sum_{m \geq 0} (-1)^m f_m \lambda^m.
\end{aligned}
\tag{A.25}
$$

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
