# Peer review of "Trans-series from condensates"

_SciPost Physics_

## Round 3 · Referee Report · Anonymous (Referee 1) · 2024-12-13

Strengths

1) This is a very systematic and solid elaboration regarding the so-called conspiracy between the factorial divergence of the formal perturbative series and relevant operators in OPE. 2) Verification of conspiracy is carried out in a simple 2D model (the Gross-Neveu model). 3) Presentation is pedagogical and well balanced.

Weaknesses

No obvious weaknesses

Report

I think this is a very solid and useful paper adding detail to the issue of the conspiracy between OPE operators and factorial divergence of the perturbation theory in asymptotical free 2D model.

I recommend publications

Requested changes

A couple of suggestions:

1) On page 3, in the beginning of Sec. 2, I would add a reference to asymptotic freedom (AF) in Gross-Neveu model. Remarkably, this model provided the first example of AF in QFT. It was discovered in 1959. See the following publication: A.A. Anselm, "A model of the field theory with nonvanishing renormalized charge" JETP 9(36), 608-611 (1959) [Russian original: ZhETF, 36, 863-868(19590].

2) Page 2, third paragraph. If my memory does not betray me, the statement at the end is not correct. I believe that in Reference [14] the OPE in the O(N) sigma model was constructed in the first subleasing order in 1/N and the conspiracy was explicitly demonstrated. The authors should mention this appropriately.

Recommendation

Publish (easily meets expectations and criteria for this Journal; among top 50%)

  • validity: high
  • significance: high
  • originality: high
  • clarity: top
  • formatting: good
  • grammar: good

Author:  Ramon Miravitllas  on 2025-01-31  [id 5174]

(in reply to Report 1 on 2024-12-13)

We would like to thank the referee for a detailed reading of our paper. We have incorporated his/her suggestions, as follows:

1) We added a footnote with a reference to the paper by Anselm (ref. 24) and to a paper by M. Shifman with a historical appraisal of Anselm’s work.

2) We have corrected the phrasing of the third paragraph in p. 2.

In addition to the changes suggested by the referees, we have added two additional references on recent studies of renormalons in two-dimensional models (refs. 8 and 9), and we have corrected a misprint in a previous version of eq. (4.22).

---

## Round 3 · Referee Report · Yizhuang Liu (Referee 2) · 2024-12-23

Strengths

The paper is pedagogical.

The paper demonstrate well features of short distance analysis for QFTs with marginal short distance asymptotics.

Weaknesses

The distinction between OPE in coordinate space and momentum space is not clarified.

Scheme conversion between the "SM" and $\overline{MS}$ schemes is fundamental, but it is only discussed during the calculation.

Scheme for removing the small-$y$ singularities of Borel integral is implicit in Eq.~(3.39), if the two terms are identified as "hard" and "soft".

Report

The paper investigated features of short distance (large Euclidean momentum) expansion of the fermion self-energy in 2D Gross Neveu model to order 1/N.

The expansion is first obtained by expanding the full result in the "SM" scheme, using Mellin-Barnes techniques.

It is then shown that after a scheme conversion, the expansion is consistent with short distance calculations in the $\overline{MS}$ scheme for the OPE coefficient functions, for operators up to dimension 2.

By comparing the full expansion and short distance calculation, non-perturbative numbers in the operator condensates are full determined, up to the renormalon ambiguities that conspire with Borel integrals for the coefficient functions.

The paper is therefore a good demonstration of short distance analysis for QFTs with marginal short distance asymptotics.

I recommend publications after minor modifications for conceptual clarity.

Requested changes

  1. The expansion is in (Euclidean) momentum space. The OPE (4.22) is in the coordinate space. Generally speaking, in going from coordinate space to momentum space, not all contributions survive. Terms that exist in small distance expansion, but dropped in large momentum expansion roughly correspond to the integration regions where the entire diagrams are soft, not just the "internal" lines. It is helpful to add a short discussion on this.

  2. The fact that the "SM" scheme can be multiplicatively converted to the standard short distance scheme $\overline{MS}$ is important. It is helpful to add more discussions on the scheme conversion in Sec.~(3.1), instead of only mentioning it during the calculation.

  3. Is the "SM" scheme also a short distance scheme? Namely, the $\beta$ and $\gamma$ functions of which can be calculated without massive propagators? The fact $\gamma_{SM}={\cal O}(\frac{1}{N^2})$ in Eq.~(3.17) is important, but in the cited reference (17) I could not find this result for Gross Neveu.

  4. In Eq.~(3.39), a special arrangement has been introduced to remove the small-$y$ singularities without introducing large $y$ divergences. But there can be other combinations. For example, one can also write the integrand as $e^{-\frac{y}{\lambda}}\frac{F_n(y)}{\lambda}+\bigg(e^{-\frac{y}{\lambda}}G_n(y)-\frac{r_{n,0}}{y}e^{-\frac{y}{\lambda'}}\bigg)-\bigg(H_n(y)-\frac{r_{n,0}}{y}e^{-\frac{y}{\lambda'}}\bigg)$, with another $\lambda'>0$ for the subtraction terms.

Although this will not affect the full result for $\Phi_{2n}$, if one would like to identify the "coefficient functions" and "operator condensates" separately, then the $\lambda'$ can be interpreted as defining the renormalization scale for them. One can also add and subtract to the Borel integrands arbitrary finite pieces and define "coefficient functions'' and "operator condensates" in various schemes. In particular, the $\overline{MS}$ scheme results calculated later do contain such finite pieces. They only cancel after combining with the operators condensates in the same scheme. Thus it is helpful also to add a discussion on this.

  1. Finally, there is a conceptual comment. The OPE expansion and the "trans-series expansion'' can be different . This happens if there are "genuine exponentially-small terms'' in the OPE coefficient functions that can not be calculated perturbatively. In this case, the "trans-series expansion'' corresponds to the "open-up" of the OPE expansion by treating the "genuine exponentially-small terms" and the "operator condensates" at the same footing.

Although not present in 2D Gross-Neveu at large N, it is likely the "genuine exponentially-small terms" exist in many Bosonic theories. In such cases, it is not true that exponentially-small terms in the trans-series are always due to condensates. But even in such cases, it is likely that OPE is correct. In the introduction the authors mentioned that the calculation can be used to check if OPE is correct. But strictly speaking, it checks a stronger form of OPE: not only OPE is correct, but there are no genuine exponentially-small terms in the coefficient functions . It is helpful to mention it in the introduction or conclusion.

Recommendation

Ask for minor revision

  • validity: top
  • significance: top
  • originality: high
  • clarity: high
  • formatting: excellent
  • grammar: excellent

Author:  Ramon Miravitllas  on 2025-01-31  [id 5173]

(in reply to Report 2 by Yizhuang Liu on 2024-12-23)
Category:
answer to question

We would like to thank the referee for a detailed reading of our paper. We have incorporated his suggestions, as follows:

1) We have made a short clarification concerning the OPE in position vs momentum space in p. 19, and refer to a paper by Novikov et al. (ref. 44) for a discussion on this point. Since we always work in momentum space, we have not commented further on this.

2 and 3) We have now collected and slightly enlarged the discussion on the change of scheme at the end of section 3.1. We have also pointed out that the result on the anomalous dimension of the field in 3.15 follows from the finiteness of the 1/N correction to the momentum part of the self-energy. The result on the anomalous dimension can also be found in eq. 2.20 of the paper by Campostrini and Rossi (reference 19 of our paper).

4) We would like to mention that the trans-series obtained from the large N result is unique, up to the choice of trans-series parameter (ambiguous part of the condensate) which is correlated to the choice of lateral resummation. Our goal is to reproduce this trans-series with a conventional “practical” version of the OPE. More complicated versions of the OPE involve an explicit intermediate scale, as suggested by the referee, and it would be interesting to address this, but it is beyond the scope of this paper.

5) We have added a comment in the third paragraph of the conclusions (p. 36) stating that there are no additional sources of exponentially small corrections in our calculation, in particular no instanton corrections. We also added a comment on the probable absence of large N instanton corrections and referred to an old paper of Avan and de Vega on this point.

In addition to the changes suggested by the referees, we have added two additional references on recent studies of renormalons in two-dimensional models (refs. 8 and 9), and we have corrected a misprint in a previous version of eq. (4.22).

---

## Editorial Decision

resubmitted